# Data Unlearning Beyond Uniform Forgetting via Diffusion Time and Frequency Selection

## Abstract

Data unlearning aims to remove the influence of specific training samples from a trained model without requiring full retraining. Unlike concept unlearning, data unlearning in diffusion models remains underexplored and often suffers from quality degradation or incomplete forgetting. To address this, we first observe that most existing methods attempt to unlearn the samples at all diffusion time steps equally, leading to poor-quality generation. We argue that forgetting occurs disproportionately across time and frequency, depending on the model and scenarios. By selectively focusing on specific time–frequency ranges during training, we achieve samples with higher aesthetic quality and lower noise. We validate this improvement by applying our time–frequency selective approach to diverse settings, including gradient-based and preference optimization objectives, as well as both image-level and text-to-image tasks. Finally, to evaluate both deletion and quality of unlearned data samples, we propose a simple normalized version of SSCD. Together, our analysis and methods establish a clearer understanding of the unique challenges in data unlearning for diffusion models, providing practical strategies to improve both evaluation and unlearning performance.

## 1 Introduction

The ability to remove the influence of training samples from a learned model, often referred to as *machine unlearning* (Bourtoule et al., 2021), has become increasingly important. Regulatory frameworks such as the "right to be forgotten" in the General Data Protection Regulation (GDPR) by the European Union and growing concerns about sensitive or proprietary data have created demand for methods that allow models to forget without costly retraining from scratch. Recently, with the development of generative models such as diffusion models (Ho et al., 2020), unlearning the unsafe concept or memorization has been actively explored through training-free sampling (Kim et al., 2025), output filtering (Yoon et al., 2025), and fine-tuning (Wang et al., 2025).

In machine unlearning, we can consider two major scenarios: (a) Concept or class unlearning, which refers to preventing the generation of certain types of samples (categorized in a particular concept or class) (Fan et al., 2024); and (b) Data unlearning, which focuses on removing individual samples. While extensive research has been conducted in the context of classification tasks, its extension to generative modelling, in particular when using diffusion models, remains underexplored (Alberti et al., 2025). Recent work on fine-tuning has begun to address this challenge with unlearning objectives such as gradient ascent, gradient importance sampling (Alberti et al., 2025), or preference optimization (Park et al., 2024). However, these approaches typically unlearn the samples in the forget set at all diffusion time steps equally. We demonstrate that this uniform unlearning in all time steps results in artifacts: generated images become noisier and aesthetically degraded, while forgetting is neither complete nor precise, as shown in Figure 1.

In this paper, we analyze the dynamics of data unlearning and argue that forgetting does not occur uniformly, but rather disproportionately across time and frequency domains. Diffusion models learn stage-dependent behaviour during training (Choi et al., 2022): later training steps close to Gaussian noise capture coarse semantics, while earlier steps close to data refine fine-grained details. Consequently, enforcing deletion across all steps may inadvertently erase either global features or only local details. To push this further, we investigate the application of a low-pass filter in the frequency domain, which encourages the model to unlearn only semantic contexts without deleting the

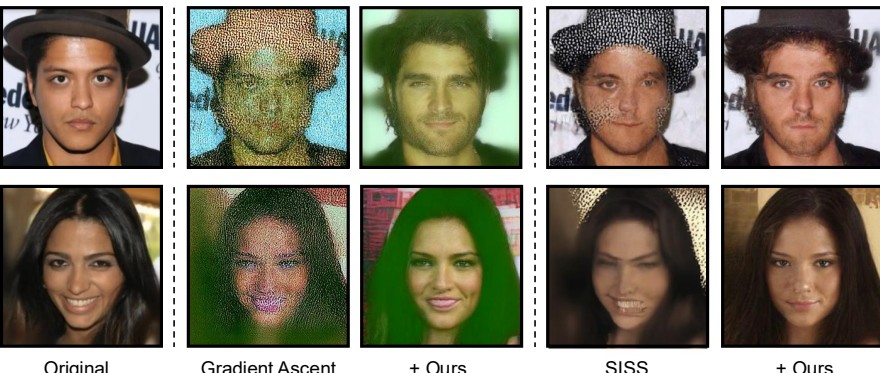

Figure 1: Illustration of quality degradation of unlearned images after data unlearning. **Letf:** two images to unlearn. **Middle:** the results of Gradient Ascent (GA, first column) and those of GA together with our approach (second column). **Right:** the results of SISS (first column) and those of SISS together with our approach (second column). Our approach, combined with existing unlearning objective functions, improves the quality of generated unlearned samples.

high-frequency components. Overall, our framework offers practical strategies for achieving effective forgetting on selective components in both time and frequency domains, while preserving the quality of unlearned samples. Our contributions are summarized as follows:

- We first observe that most existing methods attempt to unlearn the samples in the forget set at all diffusion time steps equally, leading to poor quality. We argue that forgetting occurs disproportionately across time and frequency, depending on the model and scenarios.
- We therefore introduce a novel time–frequency selective unlearning approach, which is compatible with existing unlearning objectives and effective in various experimental setups to enhance the quality of generated unlearned samples.
- We argue that current unlearning metrics, such as FID or SSCD, fail to capture the quality of unlearned samples. To address this, we introduce a simple, modified metric of SSCD.

## 2 PRELIMINARIES

### 2.1 DIFFUSION MODELS

Diffusion models (Ho et al., 2020) aim to learn a data distribution. within the diffusion process. Diffusion gradually injects noise into samples $\mathbf{x}_0$ drawn from the data distribution $p_0$, forwarding them into a fixed standard Gaussian distribution $p_T = \mathcal{N}(\mathbf{0}, \boldsymbol{I})$. The corresponding forward SDE is:

$$d\mathbf{x}_t = f(\mathbf{x}_t, t)dt + g(t)d\mathbf{w}_t, \tag{1}$$

where $f : \mathbb{R}^d \times [0, T] \to \mathbb{R}^d$ represents the drift function, $g : [0, T] \to \mathbb{R}$ denotes the diffusion coefficient, and $\mathbf{w}_t$ is the Wiener process. As diffusion models learn to reverse the above process, the reverse SDE (Anderson, 1982) is formulated as follows:

$$d\mathbf{x}_t = [f(\mathbf{x}_t, t) - g(t)^2 \nabla_{\mathbf{x}_t} \log p(\mathbf{x}_t)]dt + g(t)d\bar{\mathbf{w}}_t, \tag{2}$$

where $p(\mathbf{x}_t)$ refers to the probability density of $\mathbf{x}_t$. Diffusion models learn to match the score function $\nabla_{\mathbf{x}_t} \log p(\mathbf{x}_t)$ for denoising (Song et al., 2021). With standard DDPM forward noising process $q(\mathbf{x}_t|\mathbf{x}_0) = \mathcal{N}(\sqrt{\bar{\alpha}_t}\mathbf{x}_0, (1 - \bar{\alpha}_t)\mathbf{I})$, the conditional score can be written as $\nabla_{\mathbf{x}_t} \log q(\mathbf{x}_t|\mathbf{x}_0)$. With a weighting factor $w_t$, the training objective is then

$$\mathcal{L}_\mathcal{D}(\theta) = \mathbb{E}_{\mathbf{x}_0 \sim p_0} \mathbb{E}_{t \in [0,T]} \left[ w_t \big\| s_\theta(\mathbf{x}_t, t) - \nabla_{\mathbf{x}_t} \log q(\mathbf{x}_t|\mathbf{x}_0) \big\|_2^2 \right]. \tag{3}$$

### 2.2 DATA UNLEARNING IN DIFFUSION MODELS

**Machine Unlearning.** Suppose a model $\theta^*$ is already trained on a dataset $\mathcal{D}$. Then, our goal is to delete the influence of the forget set $\mathcal{D}_F$ from $\theta^*$, while maintaining the model utility on the retain set $\mathcal{D}_R = \mathcal{D} \setminus \mathcal{D}_F$. Specifically, our goal is to fine-tune the trained model $\theta^*$ using the forget dataset $\mathcal{D}_F$, often using the retain dataset $\mathcal{D}_R$ to mitigate quality degradation (Bourtoule et al., 2021).

**Diffusion Unlearning.** Unlearning in diffusion models can be categorized into two: concept unlearning (Gandikota et al., 2023) and data unlearning. **Concept unlearning** refers to prohibiting a diffusion model from producing images categorized in a particular type of high-level concept, including not safe for work (NSFW) material (nude and violent images, for example), or copyrighted content. Most existing work addresses concept unlearning by fine-tuning the model to suppress the desired concepts, with a focus on text-to-image tasks (Gandikota et al., 2023; Srivatsan et al., 2025), adversarial training (Zhang et al., 2024b), or preference optimization frameworks (Park et al., 2024).

Due to the high flexibility of diffusion models in controlling the generation process during sampling, unlike one-step generative models such as GANs (Goodfellow et al., 2020), training-free steering methods have been actively investigated (Singhal et al., 2025; Kim et al., 2025; Koulischer et al., 2025). These approaches leverage guidance techniques to repel the generation from specific points or embeddings, thereby achieving concept erasure without the need for fine-tuning.

On the other hand, **data unlearning** is closely related to individual data memorization and aims to remove specific data examples in accordance with the "Right to be Forgotten." For example, if a user requests the deletion of a particular face image used to train a diffusion model, the goal is to eliminate the influence of that image from the trained model. In contrast to concept erasing, for which various practical solutions have been developed, data unlearning remains relatively underexplored (Alberti et al., 2025). In general, the objective of diffusion unlearning $\mathcal{L}_{\text{UL}}(\theta)$ can be formulated as follows:

$$\min_{\theta} \mathcal{L}_{\text{UL}}(\theta) = \begin{cases} \max_{\theta} \mathcal{L}_{\mathcal{D}_{\mathcal{F}}}(\theta) = \min_{\theta} -\mathcal{L}_{\mathcal{D}_{\mathcal{F}}}(\theta), & \mathcal{D}_F \text{ only} \\ \min_{\theta} \left( -\mathcal{L}_{\mathcal{D}_{\mathcal{F}}}(\theta) + \mathcal{L}_{\mathcal{D}_{\mathcal{R}}}(\theta) \right), & \mathcal{D}_F \text{ and } \mathcal{D}_R. \end{cases} \tag{4}$$

Gradient ascent (GA) or negative gradient on forget data samples $\mathcal{D}_F$ is a key component for data unlearning (Golatkar et al., 2020). For GA, we can rewrite the loss function on the forget set as

$$\mathcal{L}_{\text{GA}}(\theta) = -\underbrace{\mathbb{E}_{\mathbf{x}_0 \sim \mathcal{D}_F} \mathbb{E}_{t \in [0,T]} \left[ w_t \big\| s_\theta(\mathbf{x}_t, t) - \nabla_{\mathbf{x}_t} \log q(\mathbf{x}_t|\mathbf{x}_0) \big\|_2^2 \right]}_{\text{Diffusion loss in Equation (3) on forget set } \mathcal{D}_F}. \tag{5}$$

EraseDiff (Wu et al., 2025) replaces the true score of the forget samples with a randomly sampled noisy image, while training normally on the retain set with scaler $\beta$:

$$\mathcal{L}_{\text{EraseDiff}}(\theta) = \mathbb{E}_{\mathbf{x}_0 \sim \mathcal{D}_F} \mathbb{E}_{\epsilon \sim \mathcal{N}(0,\mathbf{I})} \mathbb{E}_{t \in [0,T]} \left[ w_t \| s_\theta(\mathbf{x}_t, t) - (-\tfrac{1}{\sigma_t}\epsilon) \|_2^2 \right] + \beta \mathcal{L}_{\mathcal{D}_R}(\theta), \tag{6}$$

where $\epsilon$ is data-independent noise. Recently, SISS (Alberti et al., 2025) investigated the unlearning as a mixture distribution of forget and retain data with importance sampling as

$$\mathcal{L}_{\text{SISS}}(\theta) = \mathbb{E}_{\mathbf{x} \sim \mathcal{D}_F, \mathbf{x}' \sim \mathcal{D}_R} \mathbb{E}_{\mathbf{m}_t \sim q_\lambda(\cdot|\mathbf{x},\mathbf{x}')} \mathbb{E}_{t \in [0,T]} \Big[ w_{\text{keep}} \| s_\theta(\mathbf{m}_t, t) - \nabla_{\mathbf{m}_t} \log q(\mathbf{m}_t \mid \mathbf{x}') \|_2^2$$
$$- (1+\beta) w_{\text{forget}} \| s_\theta(\mathbf{m}_t, t) - \nabla_{\mathbf{m}_t} \log q(\mathbf{m}_t \mid \mathbf{x}) \|_2^2 \Big], \tag{7}$$

where $q_\lambda(\mathbf{m}_t \mid \mathbf{x}, \mathbf{x}') = (1-\lambda)q(\mathbf{m}_t \mid \mathbf{x}') + \lambda q(\mathbf{m}_t \mid \mathbf{x})$ and $w_{\text{keep}}, w_{\text{forget}}$ are importance sampling weights determined by forget and retain data, and mixture proportion $\lambda$ (e.g., $\lambda = 0.5$). In conclusion, previous methods investigate how to well-formulate Equation (4) for diffusion models.

**Quality Degradation in Unlearning.** As the goal of data unlearning is to erase information for generating a specific example, it inevitably forces the model parameters apart from a well-converged minimum. Consequently, the model also forgets previously learned features and produces lower-quality outputs. Previous research observed this quick drop in model utility when unlearning diffusion (Laria et al., 2024) or LLMs (Zhang et al., 2024a). To address the quality degradation, additional regularization of the retain set is often employed to preserve stability (Alberti et al., 2025; Wu et al., 2025). Preference optimization, such as negative preference optimization (Zhang et al., 2024a; Wang et al., 2025) or positive anchoring(Zhao et al., 2024), can be utilized for unlearning.

## 3 SELECTIVE DATA UNLEARNING IN DIFFUSION MODELS

### 3.1 UNLEARNING SCENARIOS

**Unconditional Memorization.** As illustrated in Figure 1, unconditional memorization assesses the memorization of the diffusion model to reconstruct training data without any conditional guidance. Specifically, Alberti et al. (2025) suggested noising a clean image $\mathbf{x}_0$ via the DDPM forward

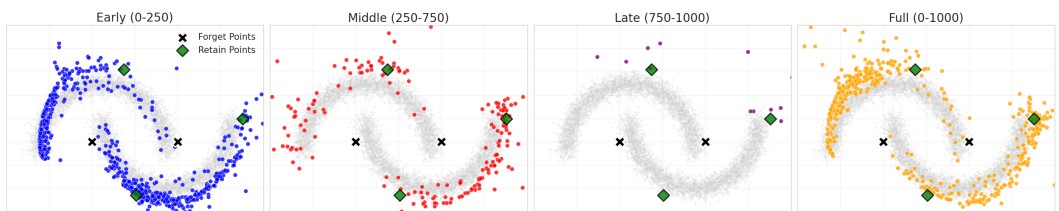

Figure 2: Illustration of unlearning the specific stage of time steps only in diffusion models.

process to an intermediate time step $t$ (e.g., $t = 250$) where the model can easily reconstruct, and measuring the similarity between the original $\mathbf{x}_0$ and the reconstructed $\hat{\mathbf{x}}_0$ from model $\theta$ with $\mathbf{x}_t$.

**Conditional Memorization.** In conditional generation, such as text embedding or class embedding, guidance causes stronger overfitting to the conditional distribution compared to the unconditional marginal distribution (Zhai et al., 2024). Therefore, conditional memorization measures how the conditional term $\epsilon_\theta(\mathbf{x}_t, e_p)$ or conditional guidance $\epsilon_\theta(\mathbf{x}_t, e_p) - \epsilon_\theta(\mathbf{x}_t, e_\emptyset)$ lead to the memorization under conditional embedding $e_p$ and null embedding $e_\emptyset$. In contrast to unconditional memorization, its evaluation starts sampling from pure noise $t = T = 1000$.

### 3.2 IMPORTANT FEATURES IN DIFFUSION UNLEARNING

The quality degradation in diffusion unlearning is shown in Figure 1. We observe that when gradient ascent is used for unlearning, the generated images after unlearning tend to lose not only the targeted information but also useful features. When combined with a retain loss, such as SISS in Equation (7), the quality is better preserved compared to using only the forget loss, but important details are still removed simultaneously. In this section, we investigate two practical solutions: (i) analysis of diffusion time stages and (ii) low-pass filtering to mitigate the quality degradation after unlearning.

**Hypothesis 1: Unlearning the specific diffusion stage is enough.** The diffusion model has its own intrinsic characteristics to use diffusion time steps $[0, T]$. Starting from Gaussian noise, diffusion models are known to learn different attributes in different time steps (Choi et al., 2022), i.e., (i) initially learn coarse features in late time steps close to time $T$, (ii) generate content during intermediate steps, and (iii) refine the details for convergence in initial time steps near time 0. However, a practical analysis of which time steps are most crucial for unlearning is missing based on fine-tuning methods. Only Zhang et al. (2025) investigated a time-aware unlearning framework, uncovering that in T2I concept unlearning, a wider range of diffusion time steps is required to fine-tune as the target concept becomes more abstract (i.e., from instance to style, class, and NSFW).

To clarify the effect of unlearning each stage, we conduct a toy experiment with a shallow diffusion model on a two half-moon dataset, visualized in Figure 2. We only split diffusion time steps for unlearning while using the same forget and retain data samples with Equation (5). Our observations are as follows: deleting only the early phase cannot prevent the model from generating forget points. On the other hand, unlearning the middle phase is most aligned with the data level, balancing the repelling of the forget dataset and the maintenance of the data manifold. Targeting the late phase is a powerful way to delete the forget samples, but also delete the distributional properties, only generating some samples near specific retain data used in unlearning fine-tuning.

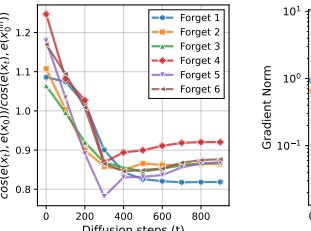
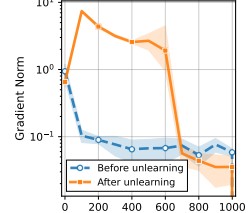

(a) Embedding similarity of $\mathbf{x}_t$ w.r.t. $\mathbf{x}_0$ and $\mathbf{x}_0^{nn}$

(b) Gradient norm of forget set before/after unlearning

Figure 3: Analysis on diffusion stage.

As shown in Figure 3 (a), when we calculate the embedding similarity on DINOv3 (Siméoni et al., 2025) of the noisy data $\mathbf{x}_t$ between other clean data samples, the similarity towards the original data $\mathbf{x}_0$ is drastically increased in the refinement stage, compared to other nearest samples. In Figure 3 (b), to compare the actual changes after standard unlearning with uniform time steps in the image-

level unlearning with CelebA-HQ, we unlearn a pre-trained model with SISS. Then, we measure the gradient norm $\|\nabla_\theta \mathcal{L}(\mathbf{x}; \theta)\|_2^2$, which reflects the magnitude of the update required to forget a sample $\mathbf{x}$ (Paul et al., 2021; Pal et al., 2025), allowing us to identify which stages are most affected by the unlearning. In diffusion training, Jain et al. (2025) utilizes gradient norm and its variance to identify important timesteps. Interestingly, the gap in the gradient norm between the model before and after unlearning is largest in the middle phase, justifying a selective unlearning with a specific window. Therefore, as the early diffusion stage mainly performs refinement, unlearning early time steps results in forgetting the details rather than deleting the memorization of a data point. These two experiments tell us that unlearning requires a selective search on time steps.

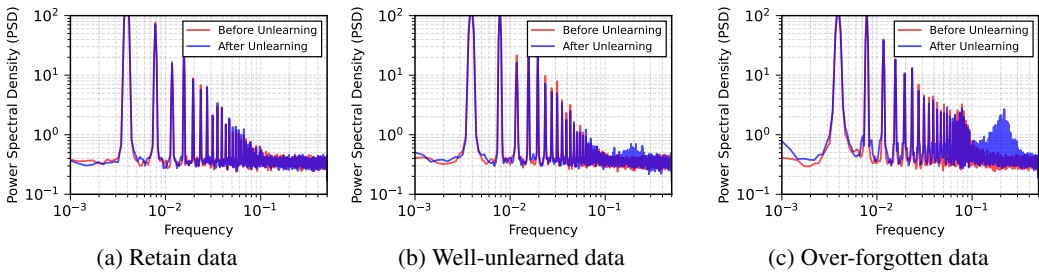

(a) Retain data  (b) Well-unlearned data  (c) Over-forgotten data

Figure 4: Power Spectral Density (PSD) before and after unlearning. For over-forgotten data, the difference in PSD is significant in high-frequency regions.

**Hypothesis 2: High-frequency components are not necessarily unlearned.** Based on the previous section, we aim to prevent the model from removing fine-grained components. We further test whether applying a low-pass filter provides selective target features for unlearning.

To clarify the differences observed during unlearning, we visualize the power spectral density, which quantifies the distribution of signal energy across frequencies, in Figure 4. For the retain dataset, the frequency domain shows no noticeable difference before and after unlearning, consistent with the results in the image space. The well-unlearned data, which indicates not losing fine-grained details while unlearning successfully, also shows small changes in high frequencies. However, over-forgotten data, which loses details during unlearning, show a much larger difference in high frequency, providing evidence for why these data yield lower-quality images. In the Section 4.4, we will further discuss these failure cases.

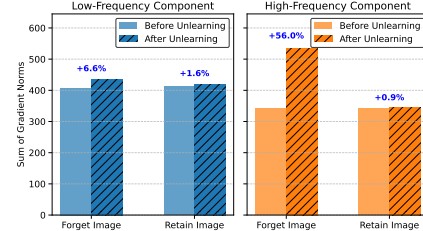

Figure 5: Gradient norm of low and high frequency parts in unlearning.

We also evaluate the gradient norm of each high and low frequency with a threshold of $10^{-1}$ in Figure 5. The change in the high-frequency region is significant in forget images, highlighting the utility of a low-pass filter for unlearning to filter out the fine-grained details.

### 3.3 SELECTIVE UNLEARNING WITH TIME STEP SELECTION AND FREQUENCY FILTERING

Based on the two observations, we propose a simple framework for selective forgetting as follows:

$$\mathcal{L}(\theta) = \underbrace{\mathbb{E}_{\tilde{\mathbf{x}}_t = \mathcal{F}(\mathbf{x}_t), \mathbf{x}_0 \sim p_0}}_{\text{Low-pass filter}} \underbrace{\mathbb{E}_{t \sim P(t)}}_{\text{Time selection}} \left[ w_t \| s_\theta(\tilde{\mathbf{x}}_t, t) - \nabla_{\tilde{\mathbf{x}}_t} \log q(\tilde{\mathbf{x}}_t | \mathbf{x}_0) \|^2 \right]. \tag{8}$$

Here, $P(t)$ assigns a non-uniform time step weighting and $\tilde{\mathbf{x}}_t$ denotes the data after applying the low-pass filter $\mathcal{F}(\cdot)$. In detail, we utilize the time scales that give a higher probability to a specific interval $t_1 \leq t \leq t_2$ for the forget data samples as follows:

$$P(t) = \begin{cases} 1 - k/(t_2 - t_1), & t_1 \leq t \leq t_2, \\ k/(T - (t_2 - t_1)), & \text{otherwise,} \end{cases} \tag{9}$$

where $0 \leq k \leq 1$ is the suppression intensity and $t_1, t_2 \in (0, T]$ with $t_1 < t_2$.

For the low-pass filter, we employ the discrete Fast Fourier Transform (FFT) as follows:

$$\mathcal{T}(u,v) = \text{FFT}(\mathbf{x}) = \sum_{x=1}^{H}\sum_{y=1}^{W}\mathbf{x}(x,y)e^{-j2\pi\left(\frac{u}{H}x + \frac{v}{W}y\right)}, \tag{10}$$

where $\mathbf{x}(x,y)$ is the pixel intensity at position $(x,y)$, $\mathcal{T}(u,v)$ is the complex coefficient at frequency $(u,v)$, and $e$ and $j$ represent Euler's constant and the imaginary unit, respectively. Its inverse is

$$\mathbf{x}(x,y) = \text{IFFT}(\mathbf{x}) = \frac{1}{HW}\sum_{u=1}^{H}\sum_{v=1}^{W}\mathcal{T}(u,v)e^{j2\pi\left(\frac{u}{H}x + \frac{v}{W}y\right)}. \tag{11}$$

To remove only the high-frequency components, we apply a masking function to the FFT results and then reconstruct the image using the inverse FFT as follows:

$$\mathcal{F}(\mathbf{x}_{i,t}) = \text{IFFT}\left(\text{FFT}(\mathbf{x}_{i,t}) \odot \beta_{i,t}(r)\right), \tag{12}$$

where $\beta_{i,t}(r) = s$ if the radius $r > r_t$, 1 otherwise. $0 \leq s \leq 1$ is a weight of high frequency and $r_t$ is a radius cutoff frequency threshold.

## 3.4 THEORETICAL GROUNDS AND PRACTICAL SELECTION

Recent theoretical studies analyze the dynamics of the reverse diffusion process and identify two transition times at which semantic structure and instance-level memorization emerge (Li & Chen, 2024; Biroli et al., 2024; Sclocchi et al., 2025; Li et al., 2025). These results provide a principled basis for understanding why unlearning should target specific diffusion stages.

**Understanding Unlearning with Speciation and Collapse Time Steps.** Biroli et al. (2024) show that denoising trajectories begin to separate according to semantic classes at the *speciation time* $t_S$

$$t_S = \arg\min_t D_{\text{class}}(p_t), \tag{13}$$

where $D_{\text{class}}(p_t)$ measures how distinguishable semantic classes are at time $t$ (e.g., via embedding similarity or class-consistency scores). For $t > t_S$, trajectories behave as nearly class-agnostic noise; for $t < t_S$, they begin encoding semantic identity. Unlearning around this point, therefore, removes conditional-level information before fine-grained details are formed. When denoising processes, trajectories start to converge toward individual training samples at the *collapse time* $t_C$ defined as

$$t_C = \arg\max_t M(p_t), \tag{14}$$

where $M(p_t)$ quantifies the instance-level attraction (e.g., nearest-neighbor consistency), indicating the point where the model loses generalization capability. For $t < t_C$, a diffusion path is attracted to single data points, thus indicating memorization. Refer to Appendix A for the details.

Setting transition times $t_1 = t_C, t_2 = t_S$ into Equation (9) allows us to analyze the unlearning phases in terms of specification and collapse. The main idea is that the diffusion trajectory requires regulation before each critical time step begins. Thus, deleting time steps $t < t_C$ is inefficient, as memorization starts at $t = t_C$. Similarly, deleting high-frequency details only degrades the quality of unlearning examples, similarly as in $t < t_C$ refining fine details (Choi et al., 2022).

For **unconditional** memorization, empirical observations in Section 3.2 demonstrate that the optimal unlearning window lies between $[t_S, t_C]$: semantic identity emerges at $t_S$, preceding the start of memorization at $t_C$. In **conditional** memorization, however, guidance effects (e.g., class or text embeddings) are already present at $t = t_S$. Therefore, it is essential to control the diffusion trajectory for $t > t_S$ before specification begins. Likewise, using training-free steering methods for $t > t_S$ is sufficient in the late stages to preserve image quality while mitigating globally harmful concepts. For example, Kim et al. (2025) applied guidance within $t > 780$, while Kirchhof et al. (2025) noted that their repellency term is most active during these late steps.

**Optimal Selection of transition time $t_S, t_C$.** Determining $t_S, t_C$ is still an ongoing area of research in diffusion models. Biroli et al. (2024) proposed finding these steps using the covariance matrix's largest eigenvalue and Shannon entropy, though their analysis assumes a Gaussian mixture setting. Conversely, other studies derive these points from signal-to-noise ratio (SNR) (Choi et al., 2022), structural pruning (Yang et al., 2024), or steering methods for sampling (Kim et al., 2025; Jain et al., 2025). We summarize these analyses in Table 1, where $t_S$ and $t_C$ are approximately similar to the time steps observed in the toy example in Figure 2.

Table 1: Comparison of critical transition points in DDPM on a $T = 1000$ ($^{\dagger}$: adaptive method).

| Reference | Basis of Analysis | $t_S$ (Speciation) | $t_C$ (Collapse) |
|---|---|---|---|
| Biroli et al. (2024)$^{\dagger}$ | Spectral / Entropy | $[500, 800]$ | $[100, 250]$ |
| Choi et al. (2022) | SNR Phases $(10^{-2}, 10^0)$ | 675 | 259 |
| Yang et al. (2024) | Structure Pruning | 750 | 250 |
| Kim et al. (2025) | Conditional Sampling | 780 | – |
| Jain et al. (2025)$^{\dagger}$ | Conditional Sampling | $[600, 800]$ | – |

**Adaptive Method for Conditional Unlearning.** In conditional memorization (Wen et al., 2024; Jeon et al., 2025), guidance magnitude $\|\epsilon_\theta(\mathbf{x}_t, e_p) - \epsilon_\theta(\mathbf{x}_t, e_\emptyset)\|$ varies by each condition and the conditional term acts stronger than its unconditional counterpart (Zhai et al., 2024). Jain et al. (2025) proposed an adaptive method for finding a transition point for prompt embedding $e_p$ to find the speciation time $t_S$. This method detects the first local minimum of the guidance magnitude during denoising from $T$, as conditional and unconditional noise predictions diverge significantly in $t > t_S$.

Leveraging this idea, we can formulate an adaptive unlearning: (i) Prior to fine-tuning, we calculate the guidance magnitude $\|\epsilon_\theta(\mathbf{x}_t, e_p) - \epsilon_\theta(\mathbf{x}_t, e_\emptyset)\|$ via diffusion sampling and find the first local minimum as $t_S$. (ii) We set the unlearning window to $t_1 = t_S$ and $t_2 = T$ for each prompt, ensuring unlearning in the conditional-specific region $t > t_S$. Refer to Appendix B for the detailed algorithm.

## 4 EXPERIMENTS

### 4.1 EXPERIMENTAL SETUPS

**Datasets and Baselines.** For CelebA-HQ (Liu et al., 2015), we test the deletion of individual samples at the image level in unconditional image generation. For Stable Diffusion v1.4 (Rombach et al., 2022), we evaluated the memorization of the LAION dataset corresponding to each prompt in text-to-image generation. We compare EraseDiff (Wu et al., 2025), Gradient Ascent (GA), and SISS, under the settings of Alberti et al. (2025). We use SISS (No IS) to reproduce the results of SISS among its variants. We also evaluate preference optimization-based approaches, including Direct Preference Optimization (DPO) (Rafailov et al., 2023) and its diffusion variant (Wallace et al., 2024), as well as Kahneman-Tversky Optimization (KTO) (Ethayarajh et al., 2024) and its diffusion variant (Li et al., 2024). Further details of experimental setups are provided in the Appendix B.

**Hyperparameters.** Following Alberti et al. (2025), we use a pretrained model from the Hugging Face diffusers package. For CelebA-HQ and Stable Diffusion, we then fine-tune the model with the Adam optimizer with a batch size of 64 and 16 with a learning rate of $5 \cdot 10^{-6}$ and $10^{-5}$, respectively. To unify the training, we use epsilon-matching (Ho et al., 2020) rather than a score-based matching.

We set the time suppression intensity $k = 0$ in Equation (9) and the high-frequency suppression weight $s = 0$ in Equation (12) to simplify experimental designs. Thus, we only tune two parameter sets: the forgetting time steps $[t_1, t_2]$ and the cutoff frequency radius $r_t$ in the FFT low-pass filter. Among the various transition times, we adopt the simplified thresholds of $t_S = 750$ and $t_C = 250$.

**Measures.** The traditional evaluation of unlearning methods is two-fold: the quality of retained data samples, measured by Frechet Inception Distance (FID) (Heusel et al., 2017), and the dissimilarity of forgotten samples from the originals, measured by SSCD (Pizzi et al., 2022). However, SSCD ignores the quality of unlearned samples, e.g., a blurry image achieves a very low SSCD, though it is not a meaningful outcome. To address this, we introduce a normalized SSCD score that considers directionality by normalizing the difference between the generated image $\hat{\mathbf{x}}_0(\mathbf{x}_t, \theta)$ and the original image $\mathbf{x}_0$, onto an $\ell_2$ ball as in adversarial attacks (Madry et al., 2018):

$$\text{SSCD}^{\text{norm}} = \text{SSCD}(\mathbf{x}_0, \mathbf{x}_0 + \rho \frac{\hat{\mathbf{x}}_0(\mathbf{x}_t, \theta) - \mathbf{x}_0}{\|\hat{\mathbf{x}}_0(\mathbf{x}_t, \theta) - \mathbf{x}_0\|_2^2 + \varepsilon}). \tag{15}$$

Here, $\varepsilon$ is a small constant to avoid division by zero, and $\rho$ is the radius for the perturbation, where we use $\rho = 100$ with a normalized image with 256x256 resolution. For generated samples, we also compute sample-wise aesthetic scores using the LAION-Aesthetic V2 predictor[1] as a quality metric.

---

[1]https://github.com/christophschuhmann/improved-aesthetic-predictor

Table 2: Comparison of unlearning methods with a relative Gain. For each method, we show the baseline scores, the scores with our method applied (+ Ours), and the resulting relative gain (%) in a separate row. For the 'Gain (%)' row, a higher value always indicates better performance. Positive gains are colored in blue, and degradations are in red.

| Method | | FID-10K ↓ | Denoising from $t = 250$ | | | Denoising from $t = 500$ | | |
|---|---|---|---|---|---|---|---|---|
| | | | SSCD ↓ | SSCD$^{norm}$ ↓ | Aesthetic ↑ | SSCD ↓ | SSCD$^{norm}$ ↓ | Aesthetic ↑ |
| Pre-trained | | 30.3 | 1.257 | - | - | - | - | - |
| Naive deletion | | 19.61 | 0.874 | 0.607 | 6.118 | 0.726 | 0.641 | 6.077 |
| EraseDiff | | 117.81 | 0.133 | 0.551 | 3.702 | 0.096 | 0.517 | 4.359 |
| GA | Base | 359.79 | 0.131 | 0.548 | 3.079 | 0.002 | 0.783 | 3.470 |
| | + Ours | 375.18 | 0.113 | 0.499 | 3.699 | 0.062 | 0.600 | 4.577 |
| | *Gain (%)* | -4.28 | +13.74 | +8.94 | +20.14 | $-3 \cdot 10^3$ | +23.37 | +31.90 |
| SISS | Base | 23.42 | 0.336 | 0.430 | 4.094 | 0.299 | 0.501 | 4.845 |
| | + Ours | 23.65 | 0.345 | 0.349 | 5.520 | 0.282 | 0.399 | 6.095 |
| | *Gain (%)* | -0.98 | -2.68 | +18.84 | +34.83 | +5.69 | +20.36 | +25.80 |
| DPO | Base | 20.58 | 0.369 | 0.404 | 5.128 | 0.313 | 0.459 | 5.058 |
| | + Ours | 21.15 | 0.344 | 0.332 | 5.614 | 0.292 | 0.393 | 6.138 |
| | *Gain (%)* | -2.77 | +6.78 | +17.82 | +9.48 | +6.71 | +14.38 | +21.35 |
| KTO | Base | 23.48 | 0.363 | 0.366 | 5.355 | 0.289 | 0.442 | 5.879 |
| | + Ours | 23.58 | 0.373 | 0.340 | 5.470 | 0.280 | 0.377 | 6.274 |
| | *Gain (%)* | -0.43 | -2.75 | +7.10 | +2.15 | +3.11 | +14.71 | +6.72 |

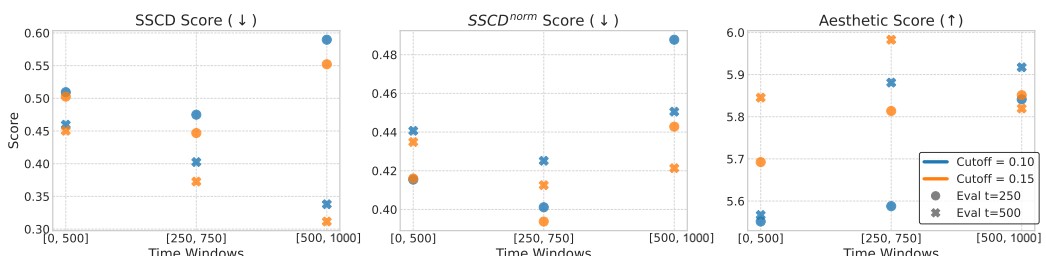

Figure 6: Ablation study with different diffusion time steps and cutoff for low-pass filter.

For Stable Diffusion, we evaluate generation quality using CLIP-IQA (Wang et al., 2023) and measure unlearning performance through the unlearning success rate, defined as the proportion of cases where all 16 memorized samples are successfully removed (Alberti et al., 2025).

## 4.2 UNCONDITIONAL DATA UNLEARNING

For CelebA-HQ, our objective is to delete six randomly sampled celebrity faces selected by (Alberti et al., 2025) from pre-trained unconditional DDPM models (Ho et al., 2020). Unlearning is applied image-by-image, and performance is evaluated after training. For SISS, DPO, and KTO, we use a single model in a continual setting, while GA erases all details after one deletion. Accordingly, we assess GA by measuring the unlearned model after each individual deletion. As explained in Section 3.1, we first inject diffusion noise to certain time steps $t = 250, 500$ and compare the reversed image using the unlearned model and the original image with SSCD score. While Alberti et al. (2025) only measured the exact memorization at $t = 250$, we also include the partial memorization at $t = 500$.

For selecting important time steps and frequency components, we first analyze whether denoising between $[t_C, t_S]$ is still practical in real images. In Figure 6, we examine an ablation study of three time ranges and FFT threshold of unlearning time steps. Similar to the toy results in Figure 2, unlearning only early steps is not effective for removing individual information; instead, it reduces the aesthetic score, resulting in noisy but still memorized samples. Therefore, we focus on the middle and middle-late intervals, where the middle steps show relatively stable forgetting and better quality preservation in image-level data unlearning. For a low-pass filter, a lower cutoff threshold effectively

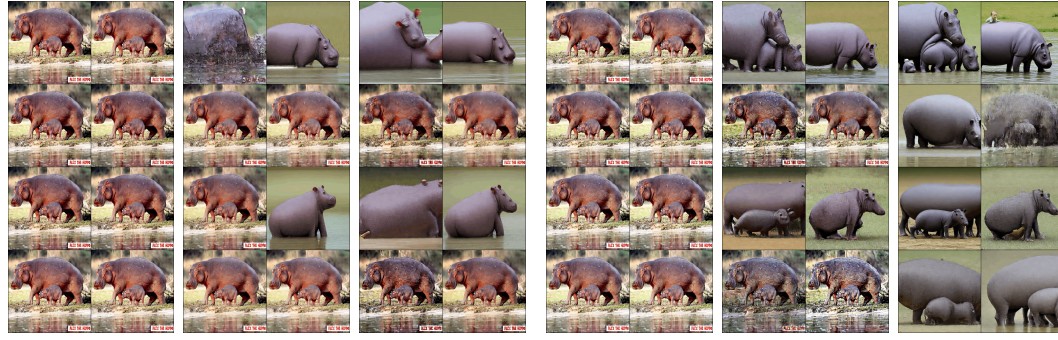

(a) Early phase of SISS (steps 0, 1, and 2)        (b) Early phase of SISS+Ours (steps 0, 1, and 2)

Figure 7: Visualization of generated images from the memorization with **fully-memorized prompt** "Mothers influence on her young hippo" as unlearning progresses. The proposed method shows faster forgetting performance while maintaining quality.

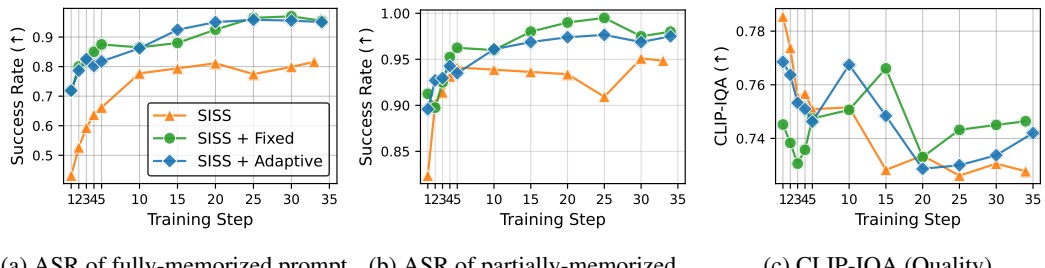

(a) ASR of fully-memorized prompt    (b) ASR of partially-memorized    (c) CLIP-IQA (Quality)

Figure 8: Unlearning success rate of 45 memorized prompts. Our methods, including fixed and adaptive, show faster unlearning convergence and high unlearning attack success rates (ASR).

reduces noisy artifacts in unlearned images. The removal of high-frequency components eliminates fragile details that are most closely tied to memorized data.

We apply our selective unlearning framework to various optimization methods (GA, SISS, DPO, and KTO), with results shown in Table 2. To ensure that a positive value consistently indicates an improvement over metric $V$, the relative gain (%) is calculated as $(V_{ours} - V_{base})/V_{base}$ for higher is better metric and $(V_{base} - V_{ours})/V_{base}$ for lower is better metric. The results indicate that both time and frequency selection achieve forgetting and preserve the quality of unlearned data examples. Overall, adding our method results in only a minor increase in FID for retained data samples, while significantly improving image quality, as indicated by the substantial gains in Aesthetic scores. Our framework demonstrates effective unlearning; at $t = 250$, it generally maintains the raw SSCD while improving $SSCD^{norm}$, and at $t = 500$, it improves both metrics compared to base models.

The comparison between SSCD and $SSCD_{norm}$ highlights the limitations of using raw similarity to evaluate unlearning. For instance, deletion-focused methods like EraseDiff and GA achieve low SSCD scores. While the results are well-aligned with target forgetting, it is merely an artifact of severe image quality degradation. Their high $SSCD_{norm}$ scores correctly reveal that their perturbation direction is ineffective for true unlearning. In contrast, our selective framework demonstrates a superior unlearning direction by achieving significant $SSCD_{norm}$ gains without this quality collapse.

### 4.3 CONDITIONAL TEXT-TO-IMAGE DATA UNLEARNING

For Stable Diffusion, we tested our strength on 45 memorized prompts within a specific target prompts selected by Alberti et al. (2025) from the LAION datasets. Fully-memorized prompts consistently generate exact replicas of the training images, whereas partially-memorized prompts represent milder versions of this phenomenon, making them relatively easier to unlearn.

For conditional cases, as explained in Section 3.4, we check to unlearn only time steps $t > t_S$, utilizing the condition on a text embedding of "memorized prompt" based rather than unconditional

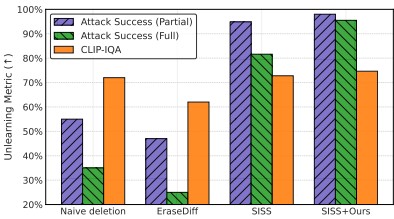

Figure 9: Comparison of unlearning methods in Stable Diffusion.

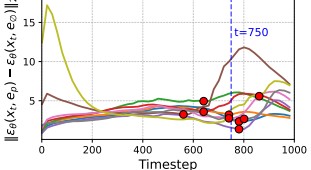

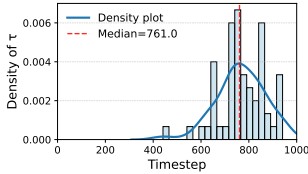

(a) Gradient norm trajectory of each conditional guidance

(b) Distribution of searched $\tau = t_S$ using adaptive selection

Figure 10: Analysis of the adaptive selection strategy.

memorization. Our experimental results, which assess both unlearning efficacy and image quality, are presented in Figures 8 and 9. As shown in Figure 8, our method achieves a higher unlearning success rate and converges faster than the baseline SISS throughout the training process. Although the CLIP-IQA score temporarily decreases during the initial phase of rapid convergence, it remains at a high level overall. The final comparison in Figure 9 further confirms that our approach yields superior unlearning (lower attack success) and significantly better image quality (higher CLIP-IQA). These observations suggest that applying unlearning uniformly across all time steps and frequency components is indeed ineffective for unlearning.

We assessed the alignment of the adaptive method proposed in Section 3.4 with our fixed time interval of $t_S = 750$ in Figure 10. Although the optimal threshold $t_S$ exhibits prompt-dependent variability, the median threshold of $t = 761.1$ corroborates that our fixed setting accurately encapsulates the critical transition point for the majority of instances. As illustrated in Figure 8, the adaptive method achieves performance similar to the fixed window in T2I tasks. However, the adaptive method introduces computational overhead by requiring a pre-sampling step to determine $t_S$. Furthermore, the outliers can have extreme windows (e.g., narrow $t_S = 900$). Therefore, we mainly adopt the fixed interval in this work, leaving the adaptive method for future exploration.

### 4.4 ADDITIONAL EXPERIMENTS

**Computation.** As shown in Table 3 using Stable Diffusion, our method requires an additional computational cost for FFT calculations of 5.99% per epoch. Additionally, the adaptive method requires a sampling step before

Table 3: Comparison of computational costs.

| Metric | SISS | Ours | Adaptive | Sampling |
|---|---|---|---|---|
| Epoch (s) | 8.51 | 9.02 | – | – |
| Total (s) | 297.68 | 315.61 | 370.66 | 55.05 |

training begins. However, since our method converges significantly faster, the actual total training time can be reduced by decreasing the number of epochs. Inference time remains unchanged.

**Notes on Failure Cases.** In the context of data unlearning, failure cases can generally be categorized into two types (Fan et al., 2025): (i) under-forgetting, where the influence of the unwanted data is not fully removed, and (ii) over-forgetting, which results in significant degradation of generation quality. Although our proposed method effectively mitigates issues from both perspectives, it is not prevent all the occasional failures. First, the PSD analysis in Figure 4c shows that over-forgetting data exhibits a distinct discrepancy in high-frequency before and after unlearning. Second, our investigation into adaptive methods reveals that the optimal effective time steps vary.

Refer to Appendix C for supplementary experiments, including different backbones, concept unlearning, and LLM evaluations, and Appendix D for visualization, including failure cases.

## 5 CONCLUSION

In this paper, we address the critical issues of quality degradation and slow convergence in data unlearning for diffusion models. Our analysis reveals that forgetting is not a uniform process, but an effective region exists across time steps and frequency domains. Based on this insight, we introduce a novel time-frequency selective framework that targets specific diffusion steps and frequency components. Our work provides a practical path toward high-fidelity machine unlearning that is compatible with existing methods. As a limitation, this work does not cover the novel design of unlearning-aware transition points search or related adaptive methods.

ETHICS STATEMENT

The primary motivation for this research is to enhance data privacy and empower individuals with control over their personal data, in alignment with regulatory principles such as the "right to be forgotten." However, we acknowledge that the unlearning process itself is not safe. The process of removing data could potentially be attacked to infer information about the unlearned information, like re-learning attacks. We recognize that robust safeguards and further research are necessary to mitigate such privacy leakages during the unlearning procedure.

REPRODUCIBILITY

Please refer to Appendix B for experimental settings. Our code is built upon the official PyTorch implementation of SISS (Alberti et al., 2025), and our source code will be released publicly.

THE USE OF LARGE LANGUAGE MODELS (LLMS)

The authors only use LLMs for the purpose of grammar correction and code modification.

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

# A    SUPPLEMENTARY ON DIFFUSION STAGES

Several theoretical papers analyze the dynamics of the denoising (sampling) process and the existence of critical windows in which particular features of the final image emerge (Li & Chen, 2024; Biroli et al., 2024; Sclocchi et al., 2025; Li et al., 2025). Here, we describe how these papers help us establish a theoretical foundation for the time selectivity in our method and provide practical guidance on which timesteps are optimal for effective unlearning in a given dataset.

In particular, we focus on the analysis by Biroli et al. (2024), who describe the generation (denoising/sampling) process given a diffusion model in terms of two distinctive times as shown in figure 11: (a) *speciation* time; and (b) *collapse* time. In the speciation time denoted by $t_S$, the sampling trajectories *start* dividing and heading towards the distribution associated with their respective class. On the other hand, in the collapse time denoted by $t_C$, trajectories start collapsing to a point in the respective distribution. Biroli et al. (2024) provides analytic expressions for $t_S$ and $t_C$ for a given dataset.

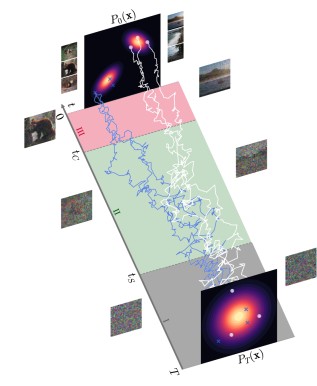

Figure 11: Dynamics stages of diffusion (Biroli et al., 2024) .

Their analysis is based on the two-Gaussian-mixture model with zero-mean and the same variance $\Delta_t$ with equal weights (which can be generalized to more Gaussians). The speciation time $t_S$ is determined by the probability that two clones (two trajectories) will go to the same Gaussian (same class) at time 0, analytically $\Lambda \exp(-2t_S) = 1$ where $\Lambda$ is the largest eigenvalue of the covariance matrix of the model. The collapse time is determined by where the (Shannon) entropy of $P_t(\boldsymbol{x}_t)$ (where $P_T(\boldsymbol{x}_T)$ is a d-dimensional isotropic Gaussian and changes during denoising) and that of a two-mixture of Gaussians match, i.e., $s(t_C) = s^{SEP}(t_C)$, where $s(t) = -\frac{1}{d} \int d\boldsymbol{x}_t P_t(\boldsymbol{x}_t) \log P_t(\boldsymbol{x}_t)$ and $s^{SEP}(t) = \frac{\log n}{d} + \frac{1}{2} + \frac{1}{2}\log(2\pi\Delta_t)$. Their explanation of why the collapse occurs where the data entropy and model entropy match is as follows: At the beginning of the denoising process $t \mapsto T$, the *excess entropy density* defined by the gap, $s^{SEP}(t) - s(t)$, is relatively small and decreases gradually during the sampling process. When $t \mapsto 0$, the model's distribution collapses to one of the modes, and this is the point where the gap becomes the largest $-\infty$. And there is a time when these two entropies perfectly match (where they define $t_C$), and after this time on, the denoising process enters the memorization phase, where the sample is attracted to a single datapoint in the training dataset.

The takeaway messages are: (i) the timesteps we find most effective for data unlearning match the theoretical speciation and collapse times, where our time-selectivity is backed up by the theory even if the theory assumes a simplistic two-Gaussian mixture data distribution; and (ii) by numerically computing $t_S, t_C$ given a dataset, one can narrow down the search space for optimal time windows for effective data unlearning.

Additionally, near the convergence in early time steps, Raya & Ambrogioni (2023) argued that data converges into a stable point or not, after a certain critical point, as shown in Figure 12. Georgiev et al. (2023) observed that the likelihood that the noisy data $\mathbf{x}_t$ is classified to a certain instance around time step $t = 650$. These findings support the existence of transition points. We clarify that Figures 12 are slight modifications of figures from (Raya & Ambrogioni, 2023) and (Georgiev et al., 2023), respectively, which we adjusted to align their time steps with our analysis.

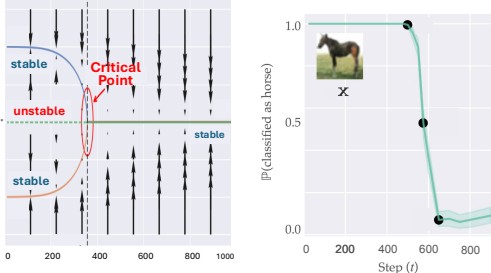

(a) Early stage (Raya & Ambrogioni, 2023)

(b) Middle-late stage (Georgiev et al., 2023)

Figure 12: Analysis on specific diffusion stages.

## B  EXPERIMENTAL DETAILS

**Experimental Setups.**    We basically follow the experimental setups in SISS (Alberti et al., 2025). We first make note of an important detail in their paper.

All the diffusion models were provided in the Huggingface `diffusers` library with a U-Net backbone. For the CelebA-HQ dataset, we used a pretrained checkpoint released by (Ho et al., 2020) at `https://huggingface.co/google/ddpm-celebahq-256`. For the Stable Diffusion experiments, we used version 1.4 at `https://huggingface.co/CompVis/stable-diffusion-v1-4` as the pretrained checkpoint and used a 50-step DDIM sampler (Song et al., 2020).

Regarding hyperparameters, we follow (Alberti et al., 2025) without additional tuning. For the proposed method, we fix the time suppression intensity at $k = 0$ in Equation (9) and the high-frequency suppression weight at $s = 0$ in Equation (12) to simplify the design. Consequently, we only adjust the key time steps $[t_1, t_2]$ with $t_1 < t_2$ by selecting two values in $[0, 250, 500, 750, 1000]$ and the cutoff frequency radius of the FFT low-pass filter, $r_t \in [0.05, 0.1, 0.15, 0.2]$. Unless otherwise specified, we set $r_t = 0.15$ as it consistently shows good performance.

For the experimental setups, we mainly use NVIDIA A40 GPUs with the PyTorch library and utilize NVIDIA A100 GPUs for parallel runs.

**Memorized Prompts of Stable Diffusion.**    Alberti et al. (2025) constructed 45 prompts in Stable Diffusion v1.4. Since only one LAION image corresponds to each prompt, synthetic datasets are generated by sampling 128 images and applying k-means clustering for classification. A "fully-memorized" prompt refers to a case where Stable Diffusion repeatedly reproduces the same outcome, whereas a "partially-memorized" prompt is obtained by manually adding or deleting tokens, producing more diverse outputs that are easier to unlearn.

**Preference Optimization.**    Direct Preference Optimization (DPO) (Rafailov et al., 2023) is widely used to evaluate preference alignment in language models. Originally developed for Reinforcement Learning from Human Feedback (RLHF), it has recently also been applied to diffusion fine-tuning (Wallace et al., 2024). For unlearning, Negative Preference Optimization (NPO) (Zhang et al., 2024a) has been proposed as an alternative to gradient descent. Unlike gradient ascent (Zhang et al., 2024a), NPO leverages the initial model as a reference point, which helps mitigate overfitting by keeping the optimization close to the initialization. In the diffusion setting, Wang et al. (2025) applied NPO for alignment. Another line of work, Kahneman–Tversky Optimization (KTO) (Ethayarajh et al., 2024), has its own strength since the method does not require positive–negative pairs. Li et al. (2024) extended KTO to diffusion for pair-free feedback alignment. In this paper, we follow the formulation of forget loss with DPO (Wallace et al., 2024) and KTO (Li et al., 2024).

To formulate this, let the per-sample diffusion loss be $\ell(\theta, \mathbf{x}) = \mathbb{E}_t[w_t \| s_\theta(\mathbf{x}_t, t) - \nabla_{\mathbf{x}_t} \log q(\mathbf{x}_t | \mathbf{x}) \|_2^2]$. We define the implicit reward as the loss reduction relative to the reference model: $r(\theta, \mathbf{x}) = \ell(\theta_{\text{ref}}, \mathbf{x}) - \ell(\theta, \mathbf{x})$. By maximizing the implicit reward gap defined by the difference in diffusion loss relative to the reference model $\theta_{\text{ref}}$, the framework effectively minimizes the likelihood of generating forget concepts while preserving general capabilities. The objectives for DPO and KTO are then given by for forget data $\mathbf{x}_f$ and retain data $\mathbf{x}_r$:

$$\mathcal{L}_{\text{DPO}}(\theta) = -\mathbb{E}_{\mathbf{x}_r \sim \mathcal{D}_R, \mathbf{x}_f \sim \mathcal{D}_F}\Big[\log \sigma\Big(\beta\big(r(\theta, \mathbf{x}_r) - r(\theta, \mathbf{x}_f)\big)\Big)\Big]. \tag{16}$$

$$\mathcal{L}_{\text{KTO}}(\theta) = \mathbb{E}_{\mathbf{x}_r \sim \mathcal{D}_R}\Big[1 - \sigma\big(\beta r(\theta, \mathbf{x}_r) - z_{\text{ref}}\big)\Big]$$
$$+ \lambda \mathbb{E}_{\mathbf{x}_f \sim \mathcal{D}_F}\Big[1 - \sigma\big(z_{\text{ref}} - \beta r(\theta, \mathbf{x}_f)\big)\Big]. \tag{17}$$

$\sigma$ is the sigmoid function, $\beta$ is a temperature hyperparameter, and $z_{\text{ref}}$ is the reference point for the reward.

**Algorithm of Adaptive Search.**    Based on Jain et al. (2025), we present a pseudocode of our adaptive method in Algorithm 1.

---

**Algorithm 1** Adaptive Transition Time Search and Selective Unlearning

---

**Require:** Pretrained model $\theta^*$; prompt embedding $e_p$; unconditional embedding $e_\emptyset$;
**Require:** Forget set $\mathcal{D}_F$; retain set $\mathcal{D}_R$; total time steps $T$
    **Phase 1: Adaptive search for transition time** $\tau$
 1: Sample $x_T \sim \mathcal{N}(0, \mathbf{I})$
 2: $\tau \leftarrow$ None
 3: $d_{\text{prev}} \leftarrow +\infty$
 4: **for** $t$ following the denoising schedule from $T$ to $1$ **do**
 5:     $d_t \leftarrow \|\epsilon_\theta(x_t, e_p) - \epsilon_\theta(x_t, e_\emptyset)\|_2$
 6:     **if** $\tau = $ None **and** $d_t > d_{\text{prev}}$ **then**
 7:         $\tau \leftarrow t_{\text{prev}}$                    ▷ first local minimum along the trajectory
 8:     **end if**
 9:     $d_{\text{prev}} \leftarrow d_t$
10:     $t_{\text{prev}} \leftarrow t$
11:     $x_{t-1} \leftarrow \text{DenoisingStep}(x_t, \epsilon_\theta)$
12: **end for**
    **Phase 2: Seletive Unlearning**
13: $t_S \leftarrow \tau$
14: Perform unlearning for timesteps $t \in [t_S, T]$

---

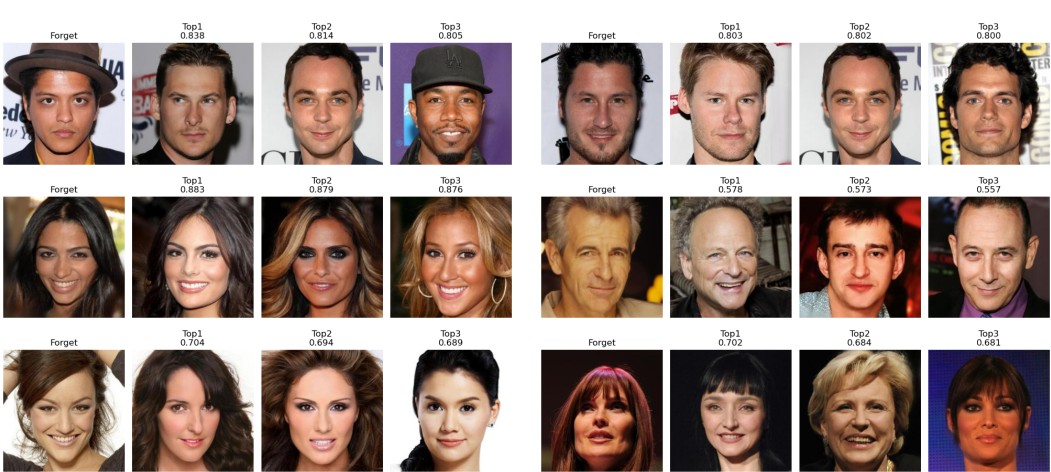

Figure 13: Forget samples and their nearest neighborhood on DINOv3 embedding (Siméoni et al., 2025).

## C SUPPLEMENTARY EXPERIMENTS

### C.1 BASE EXPERIMENTS

We illustrate some details through a toy example. For Figure 2, we construct a multi-layer perceptron diffusion model trained with the DDPM objective (Ho et al., 2020). The model is trained until convergence. After training, we delete the forget samples ($\times$) using gradient ascent, while applying gradient descent to certain retain data points ($\blacklozenge$).

Except for Figure 2, all experiments in Section 3 are conducted on the CelebHQ dataset used in the main table 2. We now provide additional results. In Figure 3, we utilize the DINOv3 embedding (Siméoni et al., 2025) to calculate the similarity between data samples. We also measured in pixel-wise similarity in Figure 14, where we failed to observe similar patterns at the embedding level. The nearest images of individual data samples are in Figure 13. Individual gradient norms are in Figure 15. Finally, we report the remaining results of Figure 4 in Figures 16 and 17.

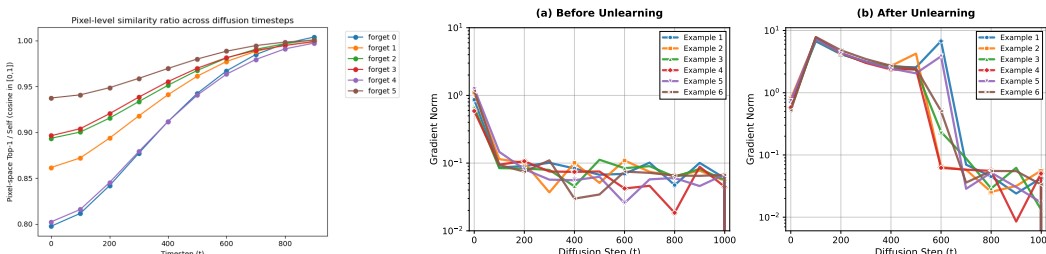

Figure 14: Similarity in pixel-level.

Figure 15: Gradient norms for each data sample before and after unlearning.

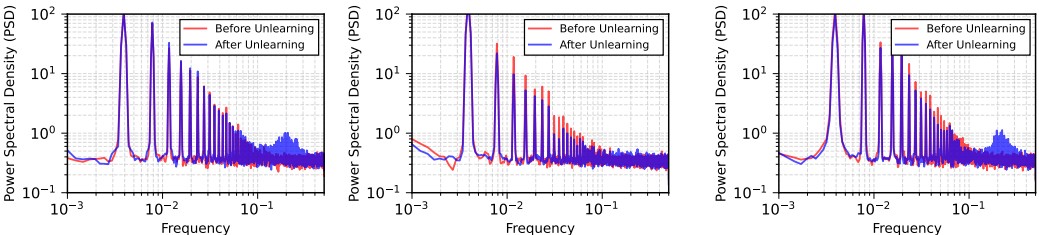

Figure 16: Additional results of Power Spectral Density (PSD) before and after unlearning on the forget dataset.

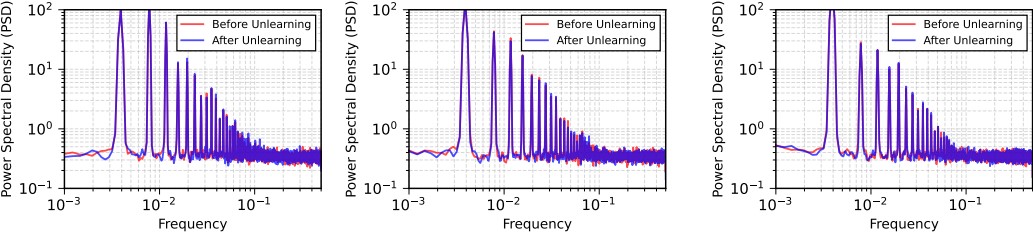

Figure 17: Additional results of Power Spectral Density (PSD) before and after unlearning on the retain dataset.

## C.2 DIFFERENT BACKBONES

To evaluate the adaptability of latent models for image-level generation, we conduct experiments using the Latent Diffusion Model (LDM) (Rombach et al., 2022) pretrained on the CelebHQ dataset[2]. The results are presented in Table 4. Furthermore, we extend our evaluation to Stable Diffusion v3.0. built upon the Diffusion Transformer (DiT) architecture (Peebles & Xie, 2023). Since Stable Diffusion v3.0 has practical guardrail methods to prevent the memorization observed in previous Stable Diffusion v1.4, we rather calculate the SSCD and SSCD$^{\text{norm}}$ while using the same prompts. The corresponding results are reported in Table 5. Tables 4 and 5 demonstrate the adaptability and effectiveness of our proposed method across diverse architectures. We did not perform additional hyperparameter tuning for these experiments.

## C.3 CONCEPT UNLEARNING

As stated in the introduction, unlearning specific concepts, such as nudity or violence, is another major direction in unlearning research. We examine whether the choice of timesteps also affects

---

[2] https://huggingface.co/CompVis/ldm-celebahq-256

Table 4: Comparison of unlearning methods on latent diffusion models with a relative Gain. For each method, we show the baseline scores, the scores with our method applied (+ Ours), and the resulting relative gain (%) in a separate row. For the 'Gain (%)' row, a higher value always indicates better performance. Positive gains are colored in blue, and degradations are in red.

| Method | | | Denoising from $t = 250$ | | | Denoising from $t = 500$ | | |
|---|---|---|---|---|---|---|---|---|
| | | FID-10K ↓ | SSCD ↓ | SSCD$^{norm}$ ↓ | Aesthetic ↑ | SSCD ↓ | SSCD$^{norm}$ ↓ | Aesthetic ↑ |
| SISS | Base | 17.800 | 0.431 | 0.547 | 4.601 | 0.281 | 0.588 | 5.092 |
| | + Ours | 16.960 | 0.437 | 0.520 | 5.076 | 0.250 | 0.536 | 4.955 |
| | *Gain (%)* | +4.72 | -1.54 | +4.95 | +10.32 | +10.90 | +8.84 | -2.69 |
| DPO | Base | 28.630 | 0.428 | 0.573 | 4.505 | 0.372 | 0.630 | 4.880 |
| | + Ours | 28.630 | 0.372 | 0.630 | 4.880 | 0.295 | 0.546 | 4.996 |
| | *Gain (%)* | 0.00 | +13.02 | -9.89 | +8.34 | +20.63 | +13.34 | +2.38 |
| KTO | Base | 28.630 | 0.372 | 0.630 | 4.880 | 0.458 | 0.472 | 5.812 |
| | + Ours | 31.360 | 0.498 | 0.576 | 5.072 | 0.457 | 0.493 | 6.446 |
| | *Gain (%)* | -9.54 | -33.65 | +8.58 | +3.93 | +0.19 | -4.48 | +10.90 |

Table 5: Quantitative comparison on Stable Diffusion v3.0. with Diffusion Transformer (DiT) backbone. Arrows indicate the direction of better performance (↑ for higher, ↓ for lower).

| Measure | SISS | SISS + Ours |
|---|---|---|
| CLIP-IQA (Partial) (↑) | 0.863 | 0.856 (-0.80%) |
| CLIP-IQA (Full) (↑) | 0.886 | 0.890 (+0.50%) |
| SSCD (Partial) (↓) | 0.349 | 0.302 (-13.67%) |
| SSCD (Full) (↓) | 0.421 | 0.362 (-14.06%) |
| SSCD Norm (Partial) (↓) | 0.704 | 0.698 (-0.87%) |
| SSCD Norm (Full) (↓) | 0.724 | 0.707 (-2.30%) |

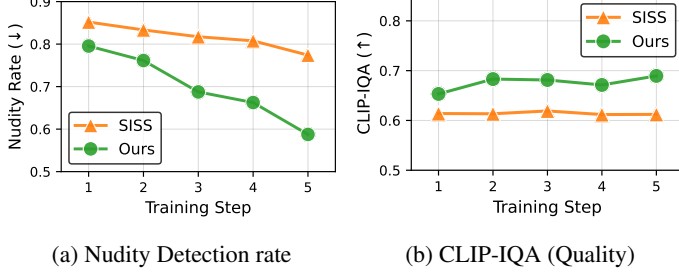

(a) Nudity Detection rate      (b) CLIP-IQA (Quality)

Figure 18: Unlearning with concept "nudity" using Ring-a-bell dataset. Our methods show faster unlearning convergence and a high nudity deletion rate.

concept generation in terms of "memorization of certain concepts." We utilize the nudity prompts used in Ring-a-bell Tsai et al. (2024) using the GitHub[3] of SAFREE (Yoon et al., 2025).

We compare SISS and SISS+Ours using the same time window used SD 1.4, and compute the CLIP-IQA quality score. For the unlearning measure, we measure the nudity detection rate using NudeNet[4], where an attack is counted if the NudeNet classifier probability exceeds 0.45. As shown in Figure 18, the proposed one shows faster convergence and superior forgetting behavior.

---

[3] https://github.com/jaehong31/SAFREE/blob/main/datasets/nudity-ring-a-bell.csv

[4] https://github.com/notAI-tech/NudeNet

Table 6: Anomaly detection scores evaluated by various LLMs on unlearned outputs. A higher score indicates the generated image is perceived as realistic (in-distribution), making it harder to detect that unlearning has occurred.

| LLM Evaluator | SISS | SISS + Ours |
|---|---|---|
| ChatGPT-5 | 0.22 | 0.67 |
| Claude Sonnet 4.5 | 0.13 | 0.83 |
| Gemini 2.5 Flash | 0.00 | 0.50 |

### C.4 IMPORTANCE OF UNLEARNED QUALITY

We argue that significant degradation or distinct artifacts in these outputs can create a privacy vulnerability. If the unlearned model generates evident anomalies for specific queries, it provides a clue to adversaries, allowing them to identify which data points were deleted (i.e., membership inference via outlier detection).

To verify this hypothesis, we conducted an experiment using advanced Large Language Models (LLMs) as anomaly detectors. We provided the LLMs with generated images and instructed them to evaluate whether the images are realistic (in-distribution) or abnormal (out-of-distribution) based on the following prompt:

> *Prompt: "You are an image quality and anomaly detection expert. You are given two image sets: The first set contains reference (in-distribution) images. Assign a score to each query image: 1.0 = Realistic / In-distribution (no visible generation problem), 0.0 = Abnormal / Out-of-distribution (clear generation failure or artifacts)."*

As shown in Table 6, baseline methods like SISS exhibited low scores, indicating that their outputs are easily distinguishable as anomalies. In contrast, our method maintains higher generation quality, effectively masking the trace of unlearning. Furthermore, recent studies (Bertran et al., 2024; Gao et al., 2025) warn that unlearning mechanisms are vulnerable against relearning attacks, which can recover deleted data using residual information such as embedding similarity. Therefore, ensuring the quality of unlearned samples is crucial not only for aesthetics but also for preventing privacy attacks.

# D    VISUALIZATION

**CelebA-HQ Results.**    We first visualize the results for each method: SISS, SISS+Ours, and variational results on different time steps in Figures 19 to 23.

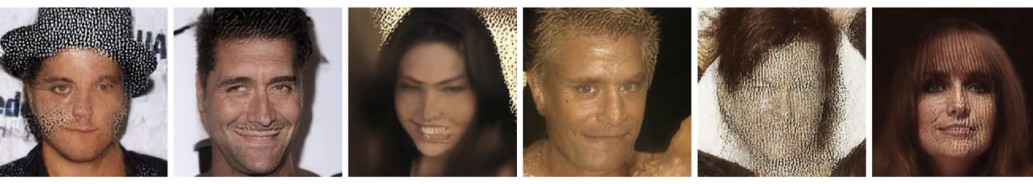

Figure 19: Unlearning with SISS.

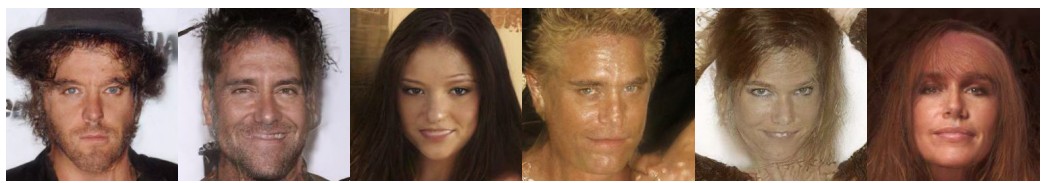

Figure 20: Unlearing with SISS+Ours.

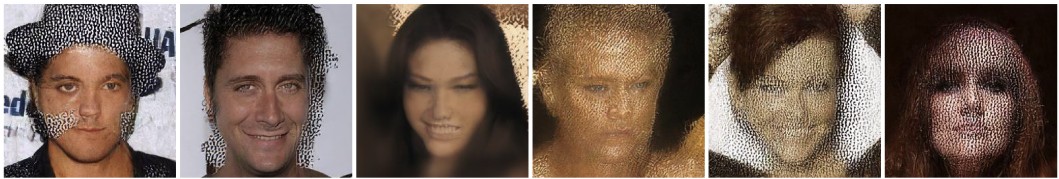

Figure 21: Unlearning on time steps [0, 500].

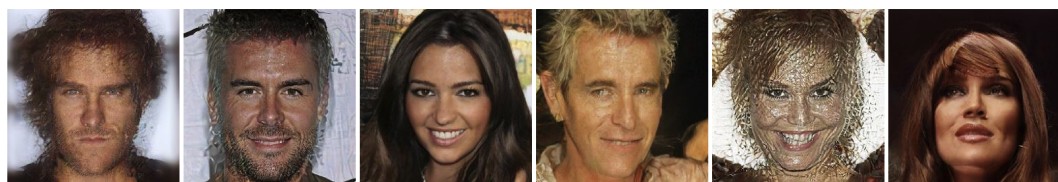

Figure 22: Unlearning on time steps [500, 1000].

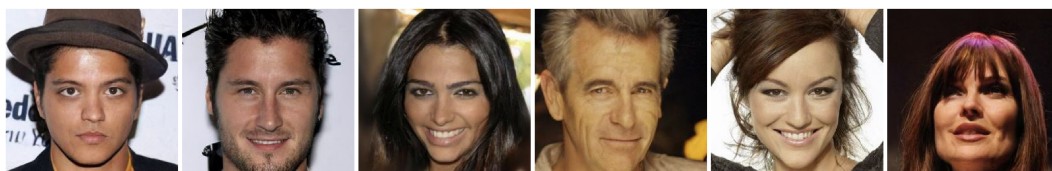

Figure 23: Unlearning on time steps [750, 1000].

**Stable Diffusion Results.** We visualize the results of text-to-image data unlearning in Figures 24 to 27.

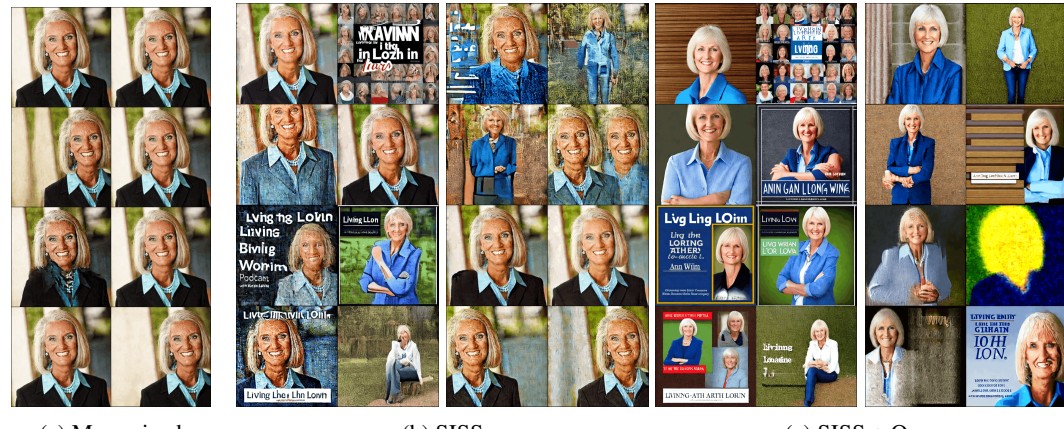

(a) Memorized          (b) SISS          (c) SISS + Ours

Figure 24: Visualization of (left) Partially-memorized and (right) Fully-memorized results after unlearning of the prompt "Living in the Light with Ann Graham Lotz".

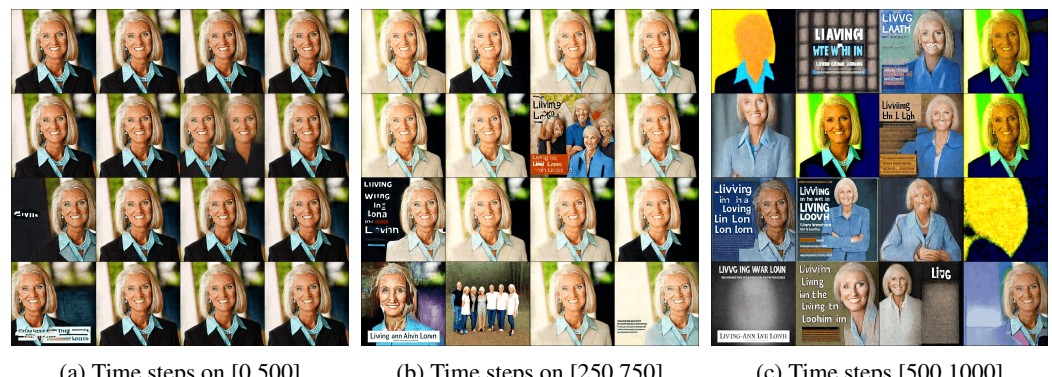

(a) Time steps on [0,500]     (b) Time steps on [250,750]     (c) Time steps [500,1000]

Figure 25: Visualization of (left) Partially-memorized and (right) Fully-memorized results after unlearning of the prompt "Living in the Light with Ann Graham Lotz" using different time steps for unlearning.

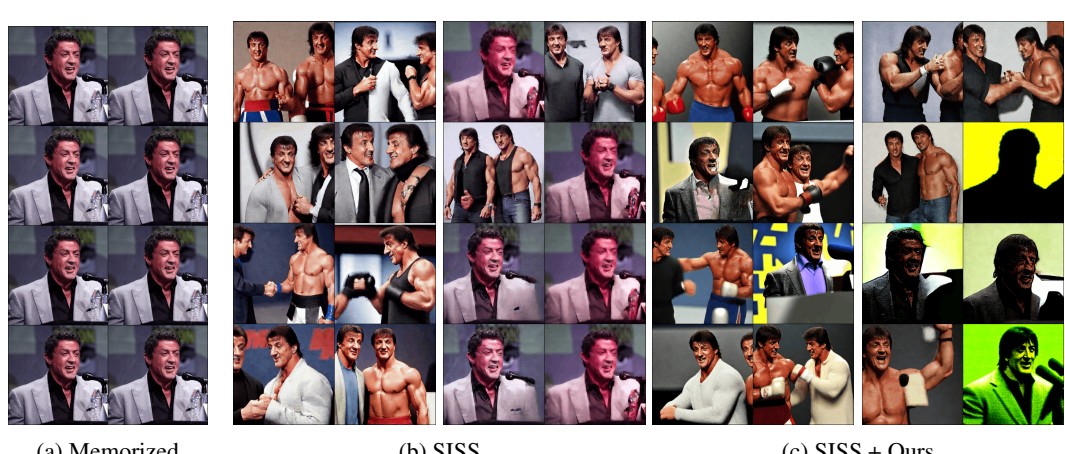

(a) Memorized          (b) SISS          (c) SISS + Ours

Figure 26: Visualization of (left) Partially-memorized and (right) Fully-memorized results after unlearning of the prompt "Rambo 5 und Rocky Spin-Off - Sylvester Stallone gibt Updates".

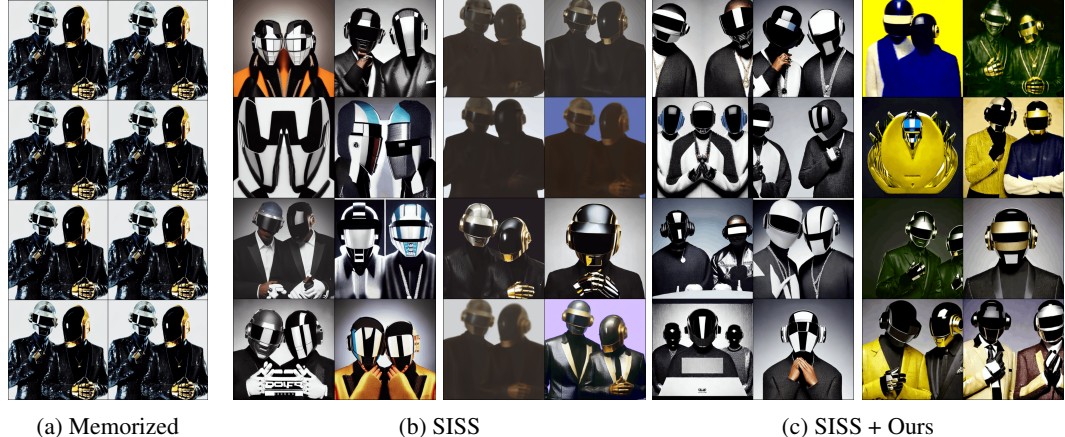

(a) Memorized          (b) SISS          (c) SISS + Ours

Figure 27: Visualization of (left) Partially-memorized and (right) Fully-memorized results after unlearning of the prompt "Daft Punk, Jay Z Collaborate on {`"Computerized"`}".

**Failure cases**    In addition to Figure 1, we will visualize some failure cases in T2I cases. Figures 28 and 29 illustrate the sampling results during unlearning SD diffusion models. In Figure 28, under-forgetting occurs, where the influence of the memorized data is not fully removed (we can also observe over-forgetting in one seed). On the other hand, in Figure 29, we can observe over-forgetting, which results in degradation of generation quality and shows meaningless yellow-blue images.

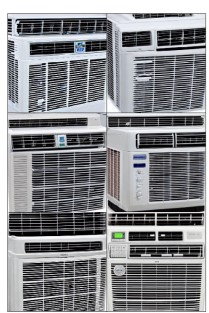 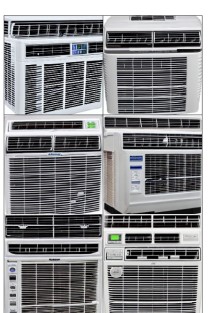 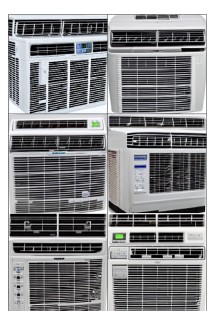 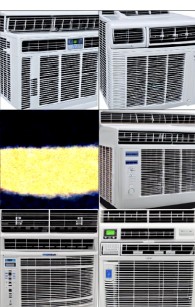

Figure 28: Under-forgetting after unlearning of the prompt " Air Conditioners & Parts".

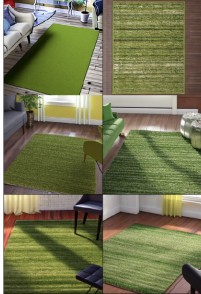 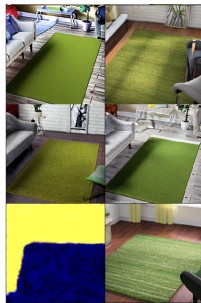 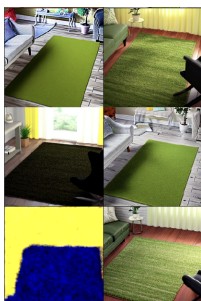 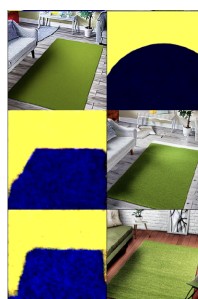

Figure 29: Over-forgetting after unlearning of the prompt "Plattville Green Area Rug by Andover Mills".

