# OpenReview forum: "Data Unlearning Beyond Uniform Forgetting via Diffusion Time and Frequency Selection"
_ICLR.cc/2026/Conference — Submitted to ICLR 2026_

### Official Review · Reviewer_EcA4 · 2025-10-29

**Soundness:** 3
**Presentation:** 2
**Contribution:** 2
**Rating:** 6
**Confidence:** 3

**Summary:**

This paper addresses data unlearning in diffusion models, which aims to remove the influence of specific training samples without full retraining. The authors identify a limitation of existing approaches: they attempt to unlearn samples uniformly across all diffusion time steps, resulting in quality degradation and incomplete forgetting. Through empirical analysis, the paper tries to demonstrate that forgetting occurs disproportionately across time and frequency domains, with different time steps encoding different levels of information (coarse semantics in late steps, fine details in early steps). Based on these insights, they propose a selective unlearning framework that applies non-uniform time step weighting and low-pass filtering to target specific time-frequency ranges during unlearning. The approach is compatible with various unlearning objectives (gradient ascent, SISS, DPO, KTO) and is validated on both image-level (CelebA-HQ) and text-to-image (Stable Diffusion) tasks, and it appears to show improved aesthetic quality and better preservation of unlearned sample quality. Additionally, they introduce SSCDnorm, a normalized version of SSCD that better captures unlearning quality by considering the directionality of changes rather than just similarity scores.

**Strengths:**

- The authors conduct a thorough empirical evaluation across multiple unlearning objectives (Gradient Ascent, SISS, DPO, KTO) and systematically explore different time step intervals and frequency cutoffs. This comprehensive analysis  demonstrates that forgetting occurs non-uniformly across specific time steps and frequency components for CelebA-HQ dataset.
- The proposed method achieves  gains in aesthetic quality (up to 34.83% improvement) while maintaining or improving unlearning performance. The results  show better preservation of visual quality in unlearned samples compared to baseline methods, addressing a critical limitation of existing approache
- The introduction of SSCDnorm is a valuable contribution that addresses limitations of standard SSCD by considering directionality of changes. This provides a more meaningful measure of unlearning quality that distinguishes between true forgetting and quality degradation.
- The paper is well-written and easy to follow

**Weaknesses:**

* The paper relies entirely on empirical observations and prior findings about image formation in diffusion processes, without providing theoretical analysis or formal justification for why selective time-frequency unlearning should work. The hypotheses are validated through experiments, but there is no principled framework to predict which time steps and frequencies are optimal for a given scenario, limiting the generalizability of the findings.
* The method requires manual tuning of time step intervals [t₁, t₂] and frequency cutoff radius r_t for each scenario. Critically, the paper shows that optimal settings differ significantly between tasks (e.g., middle steps [250, 750] for CelebA-HQ vs. late steps [750, 1000] for Stable Diffusion), but provides limited guidance on how practitioners should determine appropriate parameters for new datasets or models. The authors acknowledge this limitation in the conclusion but do not propose solutions.
* While the experiments cover multiple unlearning objectives, the evaluation is restricted to two specific settings (CelebA-HQ and Stable Diffusion v1.4 with 45 prompts). Broader evaluation across different model architectures, image resolutions, and diverse forgetting scenarios would strengthen the claims about the method's general applicability.
* While the authors acknowledge "failure cases in quality preservation" (Section 4.2), there is no systematic analysis of when and why the method fails, what characteristics of forget samples lead to poor outcomes, or how to predict and mitigate these failures.
* The "catastrophic collapse" phenomenon has also been commented on (for T2I models): "Laria, H., Gomez-Villa, A., Wang, K., Raducanu, B., & van de Weijer, J. (2024). Assessing Open-world Forgetting in Generative Image Model Customization"

**Questions:**

* Have you tested your approach on other datasets or other diffusion model architectures (e.g., DiT, different versions of Stable Diffusion)?
* Can you provide a more detailed characterization of the failure cases mentioned in Section 4.2? What percentage of forget samples experience quality degradation? Are there identifiable patterns (e.g., certain image characteristics, proximity to retain distribution) that predict failures?

---

> ### Author Response · Authors · 2025-11-22
>
> First, we thank the reviewer for raising various theoretical questions and suggesting ways to improve experimental performance. We now answer the question individually.
>
> ---
>
> **[W1, W2] Formal analysis and rationale of time step selection**
>
> We acknowledge that providing a formal grounding for our time-frequency selection strategy is crucial for delivering the core idea of our paper. We appreciate the opportunity to clarify the underlying mechanisms. Please refer to **Common response Q2. Theoretical Grounds and Practical Selection**, where we provide a rationale behind our design choices in detail and differences in evaluation schemes of image-level and T2I cases.
>
> ---
>
> **[W3, Q1] Broader Experiment**
>
> We appreciate this valuable suggestion to verify the robustness of our method. To address concerns, we applied our proposed time- and frequency-selection method to different architectures. First, we evaluated another version of Stable Diffusion 3.0, which is based on the Diffusion Transformer (DiT). Because SD 3.0 includes multiple guardrails designed to address the memorization issues observed in SD 1.4, we compute SSCD and SSCD Norm similar to the image-level experiments rather than measuring attack success rate. We use the same time window as SD 1.4. As shown below, the proposed method achieves faster memorization and forgetting compared to non-selective forgetting. Higher values indicate better performance (↑), while lower values indicate better performance (↓).
>
> | Measure | SISS | SISS + Ours |
> | --- | --- | --- |
> | CLIP-IQA (Partial) (↑)  | 0.8631 | 0.8562 (-0.80%) |
> | CLIP-IQA (Full) (↑) | 0.8857 | 0.8902 (+0.50%) |
> | SSCD (Partial) (↓) | 0.3492 | 0.3015 (-13.67%) |
> | SSCD (Full) (↓) | 0.4208 | 0.3617 (-14.06%) |
> | SSCD Norm (Partial) (↓) | 0.7044 | 0.6983 (-0.87%) |
> | SSCD Norm (Full) (↓) | 0.7236 | 0.7070 (-2.30%) |
>
> Furthermore, we evaluate the proposed method of image-level with a latent model, i.e., Latent Diffusion Model, without text-embedding pre-trained on CelebA-HQ. We use the same time window as image-level DDPM. The evaluation below shows that our method outperforms 15 metrics out of 21 metrics.
>
> | Method | FID-10K (↓) | SSCD (250) (↓) | SSCD_norm (250) (↓) | Aesthetic (250) (↑) | SSCD (500) (↓) | SSCD_norm (500) (↓) | Aesthetic (500) (↑) |
> |--------|-----------|---------------|---------------------|---------------------|----------------|----------------------|---------------------|
> | SISS      | 17.800 | 0.431 | 0.547 | 4.601 | 0.281 | 0.588 | 5.092 |
> | + Ours    | 16.960 | 0.437 | 0.520 | 5.076 | 0.250 | 0.536 | 4.955 |
> | DPO       | 28.630 | 0.428 | 0.573 | 4.505 | 0.372 | 0.630 | 4.880 |
> | + Ours    | 28.630 | 0.372 | 0.630 | 4.880 | 0.295 | 0.546 | 4.996 |
> | KTO       | 28.630 | 0.372 | 0.630 | 4.880 | 0.458 | 0.472 | 5.812 |
> | + Ours    | 31.360 | 0.498 | 0.576 | 5.072 | 0.457 | 0.493 | 6.446 |
>
> Please refer to **General Response Q1. Additional Experiments** for additional experiments in this rebuttal, including different backbones, tasks, LLM-based evaluation, and adaptive method.
>
> ---
> **[W4, Q2] Discussion of "failure cases in quality preservation"**
>
> We apologize for any confusion caused by the use of the term "failure cases" without a detailed explanation. In the context of data unlearning, failure cases can generally be categorized into two types [1]: (i) under-forgetting, where the influence of the unwanted data is not fully removed, and (ii) over-forgetting, which results in significant degradation of generation quality. Although our proposed method effectively mitigates issues from both perspectives, it does not prevent all the occasional failures.
>
> While analyzing the cause of every different failure is challenging, our analysis offers key insights into their mechanisms. First, the Power Spectral Density (PSD) analysis in Figure 4 shows that over-forgetting data exhibits a distinct discrepancy in high-frequency components before and after unlearning. Second, our investigation into adaptive methods (Common Response Q3) reveals that the optimal effective time steps vary, thus not appropriate for rare extreme cases.
>
> We move the analysis to the Experiment section and provide visualizations in the Appendix.
>
> [1] Simplicity Prevails: Rethinking Negative Preference Optimization for LLM Unlearning, NeurIPS 25.
>
> ---
>
> **[W5] Catastrophic overfitting**
>
> Thank you for clarifying the missing citation "Assessing Open-world Forgetting in Generative Image Model Customization", which explains the unintended degradation in the performance when fine-tuning, which deals with the same problems as ours. We included the paper in line 151.

---

### Official Review · Reviewer_SjNC · 2025-10-31

**Soundness:** 2
**Presentation:** 2
**Contribution:** 2
**Rating:** 4
**Confidence:** 4

**Summary:**

This work focuses on data unlearning in diffusion models. The authors identify that most existing data unlearning approaches perform uniform unlearning across all diffusion timesteps, which often causes severe degradation of image fidelity and incomplete removal of learned information. To overcome this limitation, the paper introduces a time–frequency selective unlearning framework, which applies unlearning selectively to particular middle-late timesteps and focuses on low-frequency semantic components in the latent space.  Extensive experiments on CelebA-HQ and Stable Diffusion v1.4 demonstrate that the proposed method consistently outperforms existing baselines in terms of unlearning accuracy and image quality, confirming the effectiveness of the time–frequency selective strategy.

**Strengths:**

1. The proposed time–frequency selective unlearning mechanism introduces a simple yet effective improvement over baseline approaches. This method focuses on mid-to-late timesteps and further constrains unlearning to low-frequency components through an FFT-based low-pass filter. This selective design allows the model to remove semantic information while preserving high-frequency texture details.

2. This paper introduces SSCDnorm to provide a more reliable measurement of unlearning effectiveness. While traditional SSCD often rewards methods that simply degrade image quality, SSCDnorm decouples perceptual degradation from unlearning. As a result, it can better distinguish between genuine unlearning and trivial quality loss.

3. Extensive experiments on CelebA-HQ and Stable Diffusion v1.4 validate the effectiveness of the proposed framework. On CelebA-HQ, the method achieves the lowest SSCDnorm and competitive FID and Aesthetic scores. On Stable Diffusion, selective unlearning within the [750, 1000] timestep range leads to faster convergence and a higher unlearning success rate than baselines.

**Weaknesses:**

1. The time window and frequency cutoff are chosen empirically without adaptive tuning or robustness analysis. This makes the method less stable and difficult to generalize across different models and tasks. It is recommended that the authors conduct further experiment to explore how different parameter settings affect the unlearning performance and model stability.

2. The paper lacks systematic ablation studies, and the image-level experiment [l375] only removes six faces, which is too few to convincingly verify the generality of the approach.

3. The paper reports only a simple “attack success rate” in Figure 10. Given that the selective unlearning framework may introduce potential vulnerabilities, it is recommended that the authors adopt standardized robustness evaluation metrics proposed in recent studies [1-3].



4. The proposed approach appears can be viewed as an incremental extension of existing unlearning frameworks rather than a fundamentally novel algorithm. Specifically, the method builds directly upon prior optimization-based unlearning methods such as GA and SISS, maintaining the same training objective and loss formulation. Its main modification lies in introducing time–frequency selection, which heuristically restricts the unlearning process to certain diffusion timesteps and frequency bands. While this design provides empirical improvements and clearer interpretability, it does not fundamentally alter the underlying optimization principle or introduce a new learning paradigm, thus positioning the work as an empirical extension rather than a methodological breakthrough.


[1] Yu-Lin Tsai, Chia-Yi Hsu, Chulin Xie, Chih-Hsun Lin, Jia-You Chen, Bo Li, Pin-Yu Chen,Chia-Mu Yu, and Chun-Ying Huang. Ring-a-bell! how reliable are concept removal methods for diffusion models? ICLR, 2024.

[2] Yijun Yang, Ruiyuan Gao, Xiaosen Wang, Nan Xu, and Qiang Xu. Mma-diffusion: Multimodal attack on diffusion models. CVPR, 2024.

[3] Yimeng Zhang, Jinghan Jia, Xin Chen, Aochuan Chen, Yihua Zhang, Jiancheng Liu, Ke Ding, and Sijia Liu. To generate or not? safety-driven unlearned diffusion models are still easy to generate unsafe images... for now. arXiv preprint, 2023

**Questions:**

Please refer to weaknesses.

---

> ### Author Response · Authors · 2025-11-22
>
> First of all, we thank the reviewer for reading our paper carefully and suggest various ways to improve the quality of our paper.  Since the weaknesses and questions below are aligned with other reviewers' concerns, we refer to the detailed answers in the Common responses. For the remaining question, we answer the question individually.
>
> ---
> **[W1-1] Formal analysis and rationale of time step selection**
>
> We acknowledge that providing a formal grounding for our time-frequency selection strategy is crucial for delivering the core idea of our paper. We appreciate the opportunity to clarify the underlying mechanisms. Please refer to **Common response Q2. Theoretical Grounds and Practical Selection**, where we provide a rationale behind our design choices in detail and differences in evaluation schemes of image-level and T2I cases.
>
> ---
>
> **[W1-2] Adaptive methods or parameter settings**
>
> We thank the reviewer for the comment, as this question can significantly improve the applicability of our method in various fields. Building on the insights from Common response Q2, we explored an adaptive search method to find optimal thresholds dynamically. The results of this investigation are detailed in **Common response Q3. Adaptive method**.
>
> ---
> **[W2] Ablation study**
>
> We appreciate this valuable suggestion to verify the robustness of our method. The objective of data unlearning is to delete an individual image, and deleting each image is treated as continual learning. Thus, we do not mean that we only test on six images, but for each unlearning task, our goal is to delete six face images. Similarly, for the text-to-image case, for each memorized prompt, a model is unlearned with a specific prompt. Therefore, the evaluation scheme is not limited, following the evaluation in [1].  Furthermore, we clarify that Figure 6 contains the results of the ablation.
>
> To verify the strength of selective unlearning in diffusion models, we test more backbones and datasets in this rebuttal period, including harmful "concept" unlearning as the reviewer asked in [W3]. We refer the reviewer to **Common response Q1. Additional experiments**. **We kindly ask the reviewer for any further ablation studies or questions about if required.
>
> [1] Data Unlearning in Diffusion Models, ICLR 25.
>
> ---
> **[W3] Harmful "concept" unlearning**
>
> As stated in the introduction, unlearning specific concepts, such as nudity or violence [1], is another major direction in unlearning research. We examine whether the choice of timesteps also affects concept generation in terms of “memorization of certain concepts.” We utilize the **nudity prompts** used in Ring-a-bell [1,2].
>
> We compare SISS and SISS+Ours using the same time window used SD 1.4, and compute the CLIP-IQA quality score. For the unlearning measure, we measure the nudity detection rate using NudeNet, where an attack is counted if the NudeNet classifier probability exceeds 0.45. As shown in the table below, the proposed one shows faster convergence and superior forgetting behavior.
>
> | **Step** | **Nudity detection rate (SISS) (↓)** | **Nudity detection rate (Ours) (↓)** | **CLIP-IQA (SISS) (↑)** | **CLIP-IQA (Ours) (↑)** |
> | --- | --- | --- | --- | --- |
> | **1** | 0.8519 ± 0.1472 | **0.7955 ± 0.1608** | 0.6139 ± 0.1198 | **0.6532 ± 0.1066** |
> | **2** | 0.8333 ± 0.1768 | **0.7614 ± 0.1973** | 0.6134 ± 0.1090 | **0.6831 ± 0.1093** |
> | **3** | 0.8173 ± 0.2128 | **0.6875 ± 0.2780** | 0.6191 ± 0.1145 | **0.6812 ± 0.1140** |
> | **4** | 0.8077 ± 0.2214 | **0.6625 ± 0.2890** | 0.6118 ± 0.1203 | **0.6713 ± 0.1390** |
> | **5** | 0.7740 ± 0.2599 | **0.5875 ± 0.3121** | 0.6122 ± 0.1244 | **0.6894 ± 0.1288** |
>
> [1] Ring-A-Bell! How Reliable are Concept Removal Methods For Diffusion Models?, ICLR 24.\
> [2] SAFREE: Training-Free and Adaptive Guard for Safe Text-to-Image And Video Generation, ICLR 25.

---

### Official Review · Reviewer_53d3 · 2025-11-02

**Soundness:** 3
**Presentation:** 3
**Contribution:** 2
**Rating:** 4
**Confidence:** 4

**Summary:**

The paper experiments with selective timestep and frequency selection for data unlearning in diffusion models. The authors find that careful selection of timestep and using a low-pass filter helps improve the sample quality of the unlearned images on normalized SSCD and aesthetic scores.

**Strengths:**

- A good deal of quantitative and qualitative empirical evidence is provided to justify claims that the middle range of timesteps are most effective to target and that awkward artifacts in the samples of unlearned images using existing methods are primarily the result of high frequency components being targeted.
- Analysis of the performance of preference optimization methods such as DPO and KTO are provided as well and appear to be quite effective.

**Weaknesses:**

- I would say the primary weakness of the paper is why the quality of the unlearned samples matters if they have already been successfully unlearned
- Section 3 is missing some details - see questions

**Questions:**

- Regarding section 3:
    - In Figure 4b, which loss function is being used - I'm assuming gradient ascent? Why is the gap in gradient norm before and after unlearning meaningful?
   - In Hypothesis 2, it appears there is a distinct between collapsed forget data and non-collapsed forget data. What does collapsed mean in this context?

- I may have missed it, but details surrounding the construction of the DPO and KTO preference optimization datasets would be appreciated
- Noting some typos:
    - Line 306: Citation incorrectly inside parantheticals
    - Line 318: Hyperparameter is misspelled, citation is incorrectly inside parantheticals

---

> ### Author Response · Authors · 2025-11-22
>
> First of all, we thank the reviewer for reading our paper carefully and providing various suggestions.
>
> ---
> **[W1] Why the quality of the unlearned sample matters**
>
> We also thought similarly to the reviewer at first. However, if the generated outputs of the unlearned model exhibit differences in generation, it raises another privacy concern about giving a clue which data point is deleted. We tested a simple LLM-based outlier detection that can detect the generated outputs after unlearning, with only generated quality (a higher score means lower OOD detection).
> | LLM | SISS | SISS + Ours |
> | --- | --- | ---|
> | **ChatGPT-5** | 0.22 | 0.67 |
> | **Claude Sonnet 4.5** | 0.13 | 0.83 |
> | **Gemini 2.5 Flash** | 0.00 | 0.50 |
>
> We used the simple prompt below.
> > Prompt: You are an image quality and anomaly detection expert. You are given two image sets: The first set contains reference (in-distribution) images, representing normal, realistic outputs. The second set contains query images, which may include abnormal or failed generations. Evaluate each query image based solely on visual quality, without making any personal or identity-related judgments. Assign a score to each query image: 1.0 = Realistic / In-distribution (no visible generation problem) 0.0 = Abnormal / Out-of-distribution (clear generation failure or artifacts).
>
> Furthermore, recent works [1,2,3] argue that unlearning is easily broken by relearning attacks, which use a small portion of information to figure out unlearned data samples, such as embedding similarity. Therefore, we believe that the quality of unlearned samples still matters after unlearning.
>
> Please refer to **General Response Q1. Additional Experiments** for additional experiments in this rebuttal, including different backbones, tasks, LLM-based evaluation, and adaptive method.
>
> [1] Reconstruction Attacks on Machine Unlearning: Simple Models are Vulnerable, Neurips 24.\
> [2] Meta-Unlearning on Diffusion Models: Preventing Relearning Unlearned Concepts, ICCV 25.\
> [3] Unlearning or Obfuscating? Jogging the Memory of Unlearned LLMs via Benign Relearning, ICLR 25.
>
> ---
> **[Q1-1] Loss function in Figure 4b and gradient norm**
>
> Thank you for the question. As the reviewer assumed, we utilized gradient ascent on the forget data. We further used gradient descent on the retain loss, thus SISS. We clarified this point in line 216.
> Regarding the gradient norm, its magnitude is closely related to which elements change during training, or in our case, unlearning. Therefore, a large gap in the gradient norm before and after unlearning indicates which factors actually influence the unlearning process [1], as described in Section 3.2. We also added the discussion of [2], which utilizes gradient norm and its variance to identify important timesteps in diffusion training, into Section 3.2.
>
> [1] Deep learning on a data diet: Finding important examples early in training, NeurIPS 21.\
> [2] Classifier-free guidance inside the attraction basin may cause memorization, CVPR 25.
>
> ---
> **[Q1-2] Discussion of "collapsed" data**
>
> We apologize for any confusion caused by the use of the term "collapsed" without a detailed explanation. In the context of data unlearning, failure cases can generally be categorized into two types [1]: (i) under-forgetting, where the influence of the unwanted data is not fully removed, and (ii) over-forgetting, which results in significant degradation of generation quality. Although our proposed method effectively mitigates issues from both perspectives, it does not prevent all the occasional failures.
> In this perspective, "collapsed" data means over-forgotten data samples, such as hair or details.
>
> As other reviewers also suggested, to explain "failure cases", we will unify the term "collapsed" as "over-forgetting". We move the analysis of "failure cases" to the Experiment section and provide visualizations in the Appendix.
>
> [1] Simplicity Prevails: Rethinking Negative Preference Optimization for LLM Unlearning, NeurIPS 25.
>
> ---
> **[Q2] Details about DPO and KTO**
>
> Thank you for the comment. Due to the length limit of the main paper, we explain preference optimization methods in Appendix B. We agree that the current version only explains the concepts of DPO and KTO. Thus, we added the equation below that we used during training in Appendix B.
> $$
> \mathcal{L}_ {\text{DPO}}(\theta) = - \mathbb{E}_ {x_ r \sim \mathcal{D}_ R, x_ f \sim \mathcal{D}_ F} \Bigl[ \log \sigma \Bigl( \beta \bigl( r(\theta, x_ r) - r(\theta, x_ f) \bigr) \Bigr) \Bigr]
> $$
>
> $$
> \mathcal{L}_ {\text{KTO}}(\theta) = \mathbb{E}_ {x_ r \sim \mathcal{D}_ R} \Bigl[ 1 - \sigma \bigl( \beta r(\theta, x_ r) - z_ {\text{ref}} \bigr) \Bigr]  + \lambda \mathbb{E}_ {x_ f \sim \mathcal{D}_ F} \Bigl[ 1 - \sigma \bigl( z_ {\text{ref}} - \beta r(\theta, x_ f) \bigr) \Bigr]
> $$
>
> ---
> **[Q3] Typos.**
>
> Thank you. It is obvious typos. We corrected them and conducted a thorough proofreading of the entire manuscript to ensure a well-written paper.

---

### Official Review · Reviewer_4PVH · 2025-11-02

**Soundness:** 3
**Presentation:** 3
**Contribution:** 3
**Rating:** 6
**Confidence:** 3

**Summary:**

This paper addresses data unlearning in diffusion models, focusing on removing specific training samples without full retraining. The authors observe that existing methods uniformly apply unlearning across all diffusion timesteps, leading to quality degradation and incomplete forgetting. They propose a selective unlearning framework based on middle-to-late timestep forgetting and preserving high-frequency components through low-pass filtering to maintain fine-grained details. Their approach shows improvements in aesthetic quality while maintaining unlearning effectiveness on both CelebA-HQ and Stable Diffusion experiments.

**Strengths:**

- The paper provides empirical investigation across multiple dimensions. The toy experiment on two half-moons (Figure 2) clearly illustrates how different timestep ranges affect unlearning. The gradient norm analysis (Figure 4b) provides concrete evidence that middle-to-late timesteps are most affected by unlearning. The power spectral density analysis (Figures 5-6) offers compelling evidence for the frequency filtering hypothesis. These analyses are well-grounded in prior work on diffusion model behavior.
- The proposed framework is simple and can be integrated with existing unlearning objectives without modifications. The experiments demonstrate consistent improvements across diverse settings.
- The normalized SSCD metric addresses a genuine limitation in existing evaluation where low similarity scores can result from quality degradation rather than effective unlearning. The motivation is clear and the metric provides more meaningful evaluation of unlearning direction.
- The paper tests multiple baseline methods, providing consistent gains in aesthetic scores while maintaining or improving forgetting metrics.

**Weaknesses:**

- While the appendix mentions searching over discrete time ranges and frequency cutoffs $r_t$, this still requires manual grid search for each new scenario. The paper shows that CelebA-HQ works best with [250, 750] while Stable Diffusion requires [750, 1000], but provides no systematic approach for determining these ranges a priori beyond trial and error. Without principled guidelines or even heuristics, practitioners must conduct extensive grid search for each new task, limiting practical applicability.
- The paper provides no timing comparisons or computational overhead analysis. The FFT operations and selective timestep sampling add computational cost that is never quantified. Without wall-clock time measurements or memory usage comparisons, it is difficult to assess the practical efficiency of the method.
- While the empirical analysis is extensive, the paper lacks theoretical grounding for why these specific time-frequency regions are optimal. The hypotheses are supported by experiments and references to prior work on diffusion model behavior, but there is no formal analysis or theoretical framework. Why should frequency filtering specifically help?
- The dramatically different optimal timestep ranges between CelebA-HQ and Stable Diffusion suggest that the approach may be highly task-dependent. The explanations for these differences (Section 4.3) are post-hoc and qualitative. Without clear principles for predicting optimal ranges, users cannot confidently apply this method to new scenarios.
- While the motivation for the normalized metric is reasonable, it is introduced without validation against human perceptual judgments or established quality metrics beyond aesthetic scores. The choice of $l_2$ normalization and the specific formulation in Equation 13 appear arbitrary without ablation studies or justification.
- The paper mentions "failure cases in quality preservation" on line 429, but does not analyze them or discuss when and why the method fails.


This paper addresses an important problem with well-motivated empirical analysis and demonstrates consistent improvements across multiple settings. However, the contribution is incremental, lacking principled hyperparameter selection methods and computational cost analysis, while theoretical understanding remains limited to post-hoc explanations of empirical observations.

**Questions:**

- Have you explored adaptive or learnable selection mechanisms? Rather than manual tuning, could the optimal time-frequency regions be learned from data or estimated using gradient-based importance measures?
- Why not explore learnable frequency masks? Instead of hard cutoffs at radius $r_t$, have you considered learning which frequency components to preserve or remove during unlearning?
- What causes the failure cases you mention? Can you characterize when your method fails to preserve quality and identify patterns in these failures?
- How does the method perform on other diffusion architectures? Have you tested on DiT (Diffusion Transformers) or other recent architectures beyond UNet-based models?

---

> ### Author Response · Authors · 2025-11-22
>
> We are grateful for the reviewer’s careful and comprehensive comments.
>
> Thanks to the detailed requests about the theoretical background and empirical evaluation, we believe that our work has significantly improved.
>
> ---
>
>
> **[W1, W3] Formal analysis and rationale of time step selection**
>
> We acknowledge that providing a formal grounding for our time-frequency selection strategy is crucial for delivering the core idea of our paper. We appreciate the opportunity to clarify the underlying mechanisms. Please refer to **Common response Q2. Theoretical Grounds and Practical Selection**, where we provide a rationale behind our design choices in detail and differences in evaluation schemes of image-level and T2I cases.
>
> ---
> **[W2] Computational overhead**
>
> Thank you for raising the comparison of computational costs. It is true that the proposed method increases computational time due to the selection process. We address this regarding the T2I case in two parts. In terms of training, the time step selection adds no overhead, and the FFT filter adds a slight increase of 5.99% per epoch using *torch.fft.* package. Another computation comes from the pre-training selection stage. Compared to fixed-point methods that require no pre-calculation, adaptive searching (will be explained in [Q1] below) requires one sampling before training begins. However, since our method converges significantly faster, as shown in Figure 8, the actual total training time can be reduced by decreasing the number of epochs. Inference time remains unchanged.
>
>
> | Metric / Method       | SISS  | Ours  | Adaptive | Sampling |
> |-----------------------|-------|-------|----------------|----------|
> | One-epoch (sec)  | 8.51  | 9.02  | –              | –        |
> | Total (sec)      | 297.68 | 315.61 | 370.66         | 55.05   |
>
> ---
> **[W4] Justification of normalized SSCD score**
>
> We appreciate the reviewer’s insightful comment. The Self-Supervised Copy Detection (SSCD) [1] score is widely used in memorization tasks to calculate the similarity between two images. In detail, [1] trained a **SSCD descriptor** to enforce a uniform distribution on the hypersphere and mitigate background bias via similarity normalization.
>
> However, the raw SSCD score cannot explain the unlearning by over-forgetting. For example, if a reconstructed result is simply a black image, the SSCD score will be zero. Therefore, we measured both SSCD and aesthetic score. In this direction, we thought that we needed to measure the SSCD in the same perturbation. Inspired by adversarial attacks [2, 3] that constrain perturbations within a fixed norm ($L_2$ or $L_\infty$), we came up with the simple normalized SSCD score:
> $$\mathrm{SSCD}^\text{norm} = \mathrm{SSCD}(x_0,x_0 + \rho \frac{\hat x_0(x_t,\theta) - x_0}{\|\hat x_0(x_t,\theta) -  x_0\|_2^2+\varepsilon}).$$
> This formulation evaluates degradation along a specific direction at a radius $\rho$. Consequently, we can **not only leverage the robust SSCD descriptor to assess similarity but also measure it in a fixed magnitude of perturbation**.
>
> [1] A Self-Supervised Descriptor for Image Copy Detection, CVPR 22.\
> [2] Intriguing properties of neural networks, ICLR 14.\
> [3] Towards Deep Learning Models Resistant to Adversarial Attacks, ICLR 18.
>
> ---
> **[W5, Q3] Discussion of "failure cases in quality preservation"**
>
> We apologize for any confusion caused by the use of the term "failure cases" without a detailed explanation. In the context of data unlearning, failure cases can generally be categorized into two types [1]: (i) under-forgetting, where the influence of the unwanted data is not fully removed, and (ii) over-forgetting, which results in significant degradation of generation quality. Although our proposed method effectively mitigates issues from both perspectives, it does not prevent all the occasional failures.
>
> While analyzing the cause of every different failure is challenging, our analysis offers key insights into their mechanisms. First, the Power Spectral Density (PSD) analysis in Figure 4 shows that over-forgetting data exhibits a distinct discrepancy in high-frequency components before and after unlearning. Second, our investigation into adaptive methods (Common Response Q3) reveals that the optimal effective time steps vary, thus not appropriate for rare extreme cases.
>
> We move the analysis to the Experiment section and provide visualizations in the Appendix.
>
> [1] Simplicity Prevails: Rethinking Negative Preference Optimization for LLM Unlearning, NeurIPS 25.

---

> > ### Author Response · Authors · 2025-11-22
> >
> > **[Q1] Adaptive time selection mechanism**
> >
> > We thank the reviewer for the comment, as this question can significantly improve the applicability of our method in various fields. Building on the insights from Common response Q2, we explored an adaptive search method to find optimal thresholds dynamically. The results of this investigation are detailed in **Common response Q3. Adaptive method**.
> >
> > ---
> > **[Q2] Learnable frequency mask**
> >
> > We thank the reviewer for this insightful suggestion. While this connects to the adaptive time selection discussed in [Q1], we believe that frequency selection suggests another way to improve our analysis. However, unlike the complex dynamics of time steps, the **use of a fixed low-pass filter to exclude high-frequency components is well-founded and aligns with Signal-to-Noise Ratio (SNR) principles in diffusion models** [1, 2]. Our experiments confirm that a fixed threshold provides robust performance without the need for image-specific tuning, which is aligned with recent papers that also utilize unified thresholds for frequency filtering for diffusion in dynamic motion [3] or membership inference [4].
> >
> > While investigating adaptive frequency masks in the broader vision domain is noteworthy [5-7], we observe that existing methods typically focus on learning masks during the computationally expensive training phase. Applying such mechanisms to the efficient unlearning of specific samples would likely introduce excessive computational overhead in terms of unlearning. However, we are open to exploring this direction in future work and welcome any specific suggestions from the reviewer about this direction.
> >
> > [1] Variational Diffusion Models, NeurIPS 21.\
> > [2] Elucidating the Design Space of Diffusion-Based Generative Models, NeurIPS 22.
> >
> > [3] Enhancing Motion Dynamics of Image-to-Video Models via Adaptive Low-Pass Guidance, arXiv 25.\
> > [4] Unveiling Impact of Frequency Components on Membership Inference Attacks for Diffusion Models, arXiv 25.
> >
> > [5] Learning Frequency-aware Dynamic Network for Efficient Super-Resolution, ICCV 21.\
> > [6] Graph Convolution with Low-rank Learnable Local Filters, ICLR 21.\
> > [7] Multi-scale Residual Low-Pass Filter Network for Image Deblurring, ICCV 23.
> >
> > ---
> > **[Q4] Generalization across Different Architectures**
> >
> > We appreciate this valuable suggestion to verify the robustness of our method. To address concerns, we applied our proposed time- and frequency-selection method to different architectures. First, we evaluated another version of Stable Diffusion 3.0, which is based on the Diffusion Transformer (DiT). Because SD 3.0 includes multiple guardrails designed to address the memorization issues observed in SD 1.4, we compute SSCD and SSCD Norm similar to the image-level experiments rather than measuring attack success rate. We use the same time window as SD 1.4. As shown below, the proposed method achieves faster memorization and forgetting compared to non-selective forgetting. Higher values indicate better performance (↑), while lower values indicate better performance (↓).
> >
> > | Measure | SISS | SISS + Ours |
> > | --- | --- | --- |
> > | CLIP-IQA (Partial) (↑)  | 0.8631 | 0.8562 (-0.80%) |
> > | CLIP-IQA (Full) (↑) | 0.8857 | 0.8902 (+0.50%) |
> > | SSCD (Partial) (↓) | 0.3492 | 0.3015 (-13.67%) |
> > | SSCD (Full) (↓) | 0.4208 | 0.3617 (-14.06%) |
> > | SSCD Norm (Partial) (↓) | 0.7044 | 0.6983 (-0.87%) |
> > | SSCD Norm (Full) (↓) | 0.7236 | 0.7070 (-2.30%) |
> >
> > Furthermore, we evaluate the proposed method of image-level with a latent model, i.e., Latent Diffusion Model without text-embedding, pre-trained on CelebA-HQ. We use the same time window as image-level DDPM. The evaluation below shows that our method outperforms 15 metrics out of 21 metrics.
> >
> > | Method | FID-10K (↓) | SSCD (250) (↓) | SSCD_norm (250) (↓) | Aesthetic (250) (↑) | SSCD (500) (↓) | SSCD_norm (500) (↓) | Aesthetic (500) (↑) |
> > |--------|-----------|---------------|---------------------|---------------------|----------------|----------------------|---------------------|
> > | SISS      | 17.800 | 0.431 | 0.547 | 4.601 | 0.281 | 0.588 | 5.092 |
> > | + Ours    | 16.960 | 0.437 | 0.520 | 5.076 | 0.250 | 0.536 | 4.955 |
> > | DPO       | 28.630 | 0.428 | 0.573 | 4.505 | 0.372 | 0.630 | 4.880 |
> > | + Ours    | 28.630 | 0.372 | 0.630 | 4.880 | 0.295 | 0.546 | 4.996 |
> > | KTO       | 28.630 | 0.372 | 0.630 | 4.880 | 0.458 | 0.472 | 5.812 |
> > | + Ours    | 31.360 | 0.498 | 0.576 | 5.072 | 0.457 | 0.493 | 6.446 |
> >
> > Please refer to **General Response Q1. Additional Experiments** for additional experiments in this rebuttal, including different backbones, tasks, LLM-based evaluation, and adaptive method.

---

### Author Response · Authors · 2025-11-22
**Summary of Rebuttal**

## Summary of rebuttal

We would like to thank the editor and reviewers for their careful comments and suggestions. \
The reviewers’ comments were valuable, and we believe the revised manuscript is now significantly improved in terms of **(i) broader experimental validation** and **(ii) well-grounded theoretical and practical selection**. \
As further discussion is not allowed, we summarize the reviews from our perspective.

---
### **Strength.**
Reviewers consistently emphasized our strong empirical results: “consistent improvements” (4PVH), “quantitative and qualitative empirical evidence” (53d3), “extensive experiments” (SjNC), and “thorough empirical evaluation” (EcA4).

Furthermore, reviewers highlighted the novel but simple idea of selective unlearning: "well-grounded in prior work on diffusion model behavior" (4PVH) and "a simple yet effective improvement" (SjNC).

---
### **Questions (Concerns).**
Our rebuttal focuses on (i) additional experiments (Section 4.4, Appendix C), (ii) clarifying theoretical grounds and selection criteria (Section 3.1, 3.4, Appendix A), and (iii) our adaptive method (Section 3.4, Section 4.3). \
**Major revisions (highlighted in purple) were made on Sections 3.1, 3.4, and 4.4.** Common questions are as follows:

---
### **Q1. Additional Experiments**

1. **Different backbone models** (4PVH, EcA4): We validated our method on Stable Diffusion v3.0 (DiT) and latent diffusion for unconditional image-level unlearning, confirming model-agnostic behavior.
2. **Different unlearning tasks** (SjNC): We tested concept-level unlearning (e.g., nudity) and verified that time-step selection also governs concept memorization.
3. **Importance of unlearned quality** (53d3): We show that simple LLM-based outlier detection can flag low-quality unlearned outputs; existing methods were frequently labeled as OOD.
4. **Adaptive method** (4PVH, EcA4, SjNC): See **Q3**. Prompt-wise selection yields similar results to fixed time ranges.
5. **Computation and failure cases** (4PVH): Our FFT-based step adds marginal cost and reduces the likelihood of failure cases.

---

### **Q2. Theoretical Grounds and Practical Selection (4PVH, EcA4)**

Section 3.4 now details our **theoretical foundations and practical justification**. We rely on the established three-phase view from coarse, content, and cleaning, with two critical points $t_S$ for **semantic classes** (coarse-content) and $t_C$ for **instance-level attraction** (content-cleaning).

1. The optimal unlearning window lies just **before memorization begins**. For **unconditional** memorization, the optimal unlearning window lies between $[t_S, t_C]$. Otherwise, **conditional** memorization focuses before conditional semantics emerge $t>t_S$, similar to training-free diffusion concept unlearning.

2. Furthermore, the **evaluation scheme** differs between reconstruction-based unconditional memorization and prompt-driven conditional memorization, detailed in Section 3.1.

3. For practical time selection, we provide various previous works for measuring time for $t_S$  and $t_C$ in **Table 1**.

---

### **Q3. Adaptive method (4PVH, EcA4, SjNC)**

Using the guidance magnitude $||\epsilon_\theta(x_t, e_p) - \epsilon_\theta(x_t, e_\emptyset)||$, we implement adaptive selection (Section 3.4):

(i) Before fine-tuning, run a diffusion sampling to compute $\tau=\arg\min_t ||\epsilon_\theta(x_t, e_p) - \epsilon_\theta(x_t, e_\emptyset)||$.

(ii) Set the unlearning window to $t>\tau$.

Thresholds vary across prompts, but **the median $\tau=761.1$** matches our fixed $\tau=750$. Thus, while adaptive selection is promising for future schedulers, the fixed range remains effective.

---
**For additional explanations, refer to the Detailed Q2 and Q3 below, and individual reviews.**

---

> ### Author Response · Authors · 2025-11-22
> **General Response (2)**
>
> ### Detailed answer to Q2. Theoretical Grounds and Practical Selection (4PVH, EcA4)
> We **strengthened the theoretical foundations and practical justification in Section 3.4.**
>
> **1. Theoretical grounds** (4PVH, EcA4)
> Building on recent analyses of diffusion dynamics, we theoretically explain the specific time steps with speciation ($t_S$) and collapse ($t_C$) transition points, mainly explained in [1]. These two points indicate the emergence of semantic identity and instance-level memorization in the denoising process, respectively. We clarified that unlearning should be applied between these critical points of **semantic classes** and **instance-level attraction**. The main idea is that the diffusion trajectory requires regularization, such as steering or unlearning, before each critical time step begins.
>
> For **unconditional** memorization, empirical observations in Section 3.2 demonstrate that the optimal unlearning window lies between $[t_S, t_C]$: semantic identity emerges at $t_S$, preceding the start of memorization at $t_C$. In **conditional** memorization, however, guidance effects (e.g., class or text embeddings) are already present at $t=t_S$. Therefore, it is essential to control the diffusion trajectory for $t>t_S$ before specification begins. Likewise, using training-free steering methods for $t>t_S$ is sufficient in the late stages to preserve image quality while mitigating globally harmful concepts. For example, [2] applied guidance within $t>780$, while [3] noted that their repellency term is most active during these late steps.
>
> Furthermore, the **evaluation scheme** in each scenario is different, where we add the explanation in Section 3.1. Unconditional memorization measures how well a diffusion model reconstructs a training image $x_0$ from its noised version $x_t$, produced by the DDPM forward process at an intermediate step $t$. The memorization score is the similarity between $x_0$ and the reconstruction $\hat{x}_ 0$ generated from the intermediate time step $t$ (e.g., 250).
> On the other hand, conditional memorization captures overfitting driven by a prompt embedding $e_p$. The signal appears in the conditional prediction $\epsilon_\theta(x_t, e_p)$ and especially in the guidance magnitude
> $\epsilon_\theta(x_t, e_p) - \epsilon_\theta(x_t, e_\emptyset)$, which reflects how conditional trajectories diverge from unconditional ones. Unlike the unconditional case, evaluation begins from pure noise at $t=T$, where guidance fully governs the reverse process.
>
> The differences between unconditional and conditional cases are observed in other memorization works. For membership inference attack on diffusion models, unconditional models peak at $t \in [50, 250]$ [5, 6], indicating maximal memorization in this early phase. In contrast, text-based MIA uses later time steps compared to unconditional MIA, $[400, 500]$ [7].
>
> [1] Dynamical Regimes of Diffusion Models, Nature Communications 24.\
> [2] Training-Free Safe Denoisers for Safe Use of Diffusion Models, NeurIPS 25.\
> [3] Shielded Diffusion: Generating Novel and Diverse Images using Sparse Repellency, ICML 25.\
> [4] Data Unlearning in Diffusion Models, ICLR 25.
> [5] Are Diffusion Models Vulnerable to Membership Inference Attacks? ICML 23.
> [6] An Efficient Membership Inference Attack for the Diffusion Model by Proximal Initialization, ICLR 24.
> [7] Membership Inference on Text-to-image Diffusion Models via Conditional Likelihood Discrepancy, NeurIPS 24.
>
>  **2. Practical selection of time steps** (4PVH, EcA4, SjNC)
> Determining  $t_S, t_C$ is still an ongoing area of research in diffusion models. [1] proposed finding these steps using the covariance matrix's largest eigenvalue and Shannon entropy, though their analysis assumes a Gaussian mixture setting. Conversely, other studies derive these points from signal-to-noise ratio (SNR) [2], structural pruning [3], or steering methods for sampling [4,5]. We summarize these analyses in the Table below, where $t_S$ and $t_C$ are approximately similar to the time steps we used. $\dagger$ means supporting the adaptive method.
>
> **We add the detailed analysis in Section 3.4 and the theoretical parts of transition points and diffusion stages in Appendix A.**
>
> | Reference| Basis of Analysis| t_S (Speciation) | t_C (Collapse) |
> |-------------------------|-----------------------------|-------------------|----------------|
> | [1] †| Spectral / Entropy| [500, 800]| [100, 250]|
> | [2]  | SNR Phases (0.01,  1)| 675| 259|
> | [3] | Structure Pruning| 750| 250|
> | [4] | Conditional Sampling| 780| –|
> | [5]†| Conditional Sampling| [600, 800]| –|
>
> [1] Dynamical Regimes of Diffusion Models, Nature Communications 24.\
> [2] Perception prioritized training of the diffusion model, CVPR 22.\
> [3] Denoising Diffusion Step-aware Models, ICLR 24.\
> [4] Shielded Diffusion: Generating Novel and Diverse Images using Sparse Repellency, ICML 25.\
> [5] Classifier-Free Guidance inside the Attraction Basin May Cause Memorization, CVPR 25.

---

> ### Author Response · Authors · 2025-11-22
> **General Response (3)**
>
> ### Detailed answer to Q3. Adaptive method (4PVH, EcA4, SjNC)
>
> We thank the reviewers for suggesting the development of an adaptive threshold search. We draw inspiration from training-free sampling methods that analyze the divergence between conditional and null trajectories. Motivated by studies on T2I memorization [1, 2] and guidance magnitude $||\epsilon_\theta(x_t, e_p) - \epsilon_\theta(x_t, e_\emptyset)||$ for prompt embedding $e_p$, [3] proposes identifying a transition point $\tau$, which we equally treated as with the speciation time $t_S$ in our anlaysis on Q2. Theoretical Grounds and Practical Selection). Specifically, [3] detects the first local minimum of the guidance magnitude during denoising from $T$, based on the observation that the $L_2$ norms of conditional and unconditional noise predictions diverge significantly during the coarse sampling stage.
>
> Adopting this logic, **we formulated an adaptive selection mechanism to determine unlearning time steps** as follows: (i) Prior to fine-tuning, we run a diffusion sampling to calculate the magnitude $||\epsilon_\theta(x_t, e_p) - \epsilon_\theta(x_t, e_\emptyset)||$ for memorized embedding $e_p$ and identifying $\tau=t_S$. (ii) We dynamically set the unlearning window in Eq. 9 to $t_1=\tau$ and $t_2=T$, thereby restricting unlearning to $t > \tau$, which is an adaptively determined time range by each prompt.
>
> **Experimental results**: We first checked how well the adaptive method aligns with our fixed time range of $[750, 1000]$. Similar to [3], we found that the optimal threshold $\tau$ varies by prompt. Interestingly, **the median of these thresholds is $t=761.1$**, which confirms that our fixed setting effectively captures the critical transition point for the majority of cases.
>
> Regarding performance, the adaptive method yields results comparable to the fixed window. However, it introduces specific drawbacks. Unlike the fixed threshold, the method necessitates a pre-training sampling step to determine $\tau$, incurring additional computational overhead. Furthermore, the adaptive approach is sensitive to outliers, potentially yielding suboptimal windows (e.g., overly narrow ranges are determined like $[900, 1000]$) that lead to inconsistent performance. Therefore, while we believe this adaptive method holds promise for future work involving different schedulers or experimental settings, the fixed interval still remains effective.
>
> | Step | ASR_SISS (↑)  | ASR_Fixed (↑) | ASR_Adaptive (↑)|  | CLIP_SISS (↓) | CLIP_Fixed (↓)| CLIP_Adaptive (↓)|
> | --- | --- | --- | --- | --- | --- | --- | --- |
> | 1 | 0.8235 | 0.9125 | 0.8958 |  | 0.7854 | 0.7452 | 0.7685 |
> | 2 | 0.9020 | 0.8975 | 0.9271 |  | 0.7738 | 0.7383 | 0.7637 |
> | 3 | 0.9142 | 0.9250 | 0.9297 |  | 0.7554 | 0.7306 | 0.7532 |
> | 10 | 0.9387 | 0.9600 | 0.9609 |  | 0.7516 | 0.7506 | 0.7674 |
> | 20 | 0.9338 | 0.9900 | 0.9740 |  | 0.7336 | 0.7331 | 0.7287 |
> | 30 | 0.9510 | 0.9750 | 0.9688 |  | 0.7306 | 0.7450 | 0.7387 |
>
> [1] Detecting, Explaining, and Mitigating Memorization in Diffusion Models, ICLR 24.\
> [2] Understanding and Mitigating Memorization in Generative Models via Sharpness of Probability Landscapes, ICML 25.\
> [3] Dynamic Negative Guidance of Diffusion Models, ICLR 25.

---

### Meta-Review · Area_Chair_U78r · 2026-01-06

**Summary:**

**Summary:** This paper studies data unlearning in diffusion models. The authors showed 1) that unlearning at different denoising time steps leads to different unlearning behaviors, and 2) that the unlearned model often generates deviated high-frequency components for the unlearned images and hence leads to poor quality. As such, they proposed to unlearn different time steps differently and suppress the unlearning for high-frequency components. They further propose a new metric named normalized SSCD to emphasize the unlearning performance.

**Strength (by reviewers):** systematic empirical investigation; simple and compatible proposed methods; well-motivated metric; multiple baselines; notable unlearning gain; well-written paper

**Weakness (by reviewers):** lacking principled methods to determine the time step range; lacking theoretical grounding and formal analysis of the proposed method; lacking further justification on the proposed metric; lacking in-depth analysis on failure cases; the motivation to keep the quality of the unlearned examples is unclear; missing details in section 3 (e.g., why gradient norms matter); lacking ablation studies and more robust analysis; lacking computational/time analysis; the proposed method appears technically incremental (the AC disagrees with this.)

**Decision:** The paper received an average original score of 5 (4, 4, 6, 6). The authors provided extensive rebuttals (including further experiments). None of the reviewers was involved in the discussions. The AC read the paper, reviews, and rebuttals (and the revised manuscript). While the AC acknowledges that many concerns/weaknesses have been (partially) addressed, some remain unresolved. Further, the added theoretical grounding needs a significant change in sections 1 & 3 to make the paper consistent overall.

Besides, the AC agrees with the concerns raised by Reviewer 53d3.
- First, the paper should have a better motivation for why the quality of the unlearned examples matters and incorporate it in the main story. As mentioned in Section 2, prior studies mainly assess the quality using the retained dataset. More specifically, prior unlearning work focused on ensuring that the model unlearns the target examples/concepts while retaining its generative ability for the other examples/concepts. If the authors aim to assess a different aspect, it should be clearly motivated at the beginning. Please note that, according to Table 2, the proposed method does not improve but slightly degrades the quality of the retained dataset (why?); the main quality improvements are on the unlearned examples. Such a different focus should be clearly justified early in the paper.
- Second, the explanation for the meaning and use of gradient norms is still unclear. While gradient norms have been used widely in various ML studies, their prior usages were different from this paper. Typical uses of the gradient norms are *during* training, while the authors compare the gradient norms between the original and the unlearned models (it is also unclear why, after unlearning, the gradient norm increased). Further, the large gap in gradient norms was interpreted differently in the paper (thus, ambiguous). In Fig. 3 (b), the authors argued that the large gap indicates where unlearning should happen; in Fig. 5, however, the large gap indicates where unlearning should be suppressed (i.e., high-frequency part).
- Third, in Section 3, each figure should have a self-contained caption; each analysis should have a self-contained or clearly-referenced setup.

Overall, the AC finds that the paper, at its current version, lacks a coherent/consistent story. It touches a bit too many aspects---how should unlearning be evaluated, what should be the right metric, how should unlearning be implemented, why do different time steps lead to different unlearning behaviors and how to properly leverage/select them---but 1) some aspects were touched at a high level without details analyes or design choice comparison and 2) the connections across these aspects are not strong enough. The AC thus thinks the paper needs a major revision and recommends *rejection.*

**Reviewer Concerns:**

**Reviewer 4PVH.**
- **Addressed:** different architectures, frequency selection, adaptive step selection (one idea is investigated), computation analysis, failure case analysis
- **Remained:** theoretical grounding and practical selection (addressed partially), justification of the proposed metric (e.g., lacking design choice comparison)

**Reviewer 53d3.**
- **Addressed:** some details in section 3 and baselines
- **Remained:** why the quality of the unlearned sample matters (addressed partially), gradient norm (partially addressed)

**Reviewer SjNC.**
- **Addressed:** adaptive step selection (one idea is investigated), harmful concept unlearning
- **Remained:** theoretical grounding and practical selection (addressed partially), incremental extension

**Reviewer EcA4.**
- **Addressed:** broader experiments (mainly different architectures), failure case analysis
- **Remained:** theoretical grounding and practical selection (addressed partially)

**Reviewer Scores:**

**Reviewer 4PVH (6 to 6):** concerns/questions were partially addressed; the reviewer likely will keep the original score

**Reviewer 53d3 (4 to 4):** concerns/questions were partially addressed; the reviewer likely will keep the original score

**Reviewer SjNC (4 to 4):** concerns/questions were partially addressed; the reviewer likely will keep the original score. (Please note that the AC does not agree with the incremental extension concern.)

**Reviewer EcA4 (6 to 6).** concerns/questions were partially addressed; the reviewer likely will keep the original score

---

### Decision · Program_Chairs · 2026-01-26

Reject